# A pilot study of alternative substrates in the critically Ill subject using a ketogenic feed

Angela McNelly[1], Anne Langan[2], Danielle E. Bear [3,4], Alexandria Page[5], Tim Martin[5], Fatima Seidu[5], Filipa Santos[5], Kieron Rooney [6], Kaifeng Liang[1], Simon J. Heales[7], Tomas Baldwin[8], Isabelle Alldritt [9], Hannah Crossland[9], Philip J. Atherton[9], Daniel Wilkinson[9], Hugh Montgomery[10,11], John Prowle[1,5], Rupert Pearse[1,5], Simon Eaton [8] & Zudin A. Puthucheary [1,5] ✉

Bioenergetic failure caused by impaired utilisation of glucose and fatty acids contributes to organ dysfunction across multiple tissues in critical illness. Ketone bodies may form an alternative substrate source, but the feasibility and safety of inducing a ketogenic state in physiologically unstable patients is not known. Twenty-nine mechanically ventilated adults with multi-organ failure managed on intensive care units were randomised (Ketogenic $n = 14$, Control $n = 15$) into a two-centre pilot open-label trial of ketogenic versus standard enteral feeding. The primary endpoints were assessment of feasibility and safety, recruitment and retention rates and achievement of ketosis and glucose control. Ketogenic feeding was feasible, safe, well tolerated and resulted in ketosis in all patients in the intervention group, with a refusal rate of 4.1% and 82.8% retention. Patients who received ketogenic feeding had fewer hypoglycaemic events (0.0% vs. 1.6%), required less exogenous international units of insulin (0 (Interquartile range 0-16) vs.78 (Interquartile range 0-412) but had slightly more daily episodes of diarrhoea (53.5% vs. 42.9%) over the trial period. Ketogenic feeding was feasible and may be an intervention for addressing bioenergetic failure in critically ill patients. Clinical Trials.gov registration: NCT04101071.

Critical illness is a state of ill health with vital organ dysfunction and a high risk of imminent death if care (pharmacological or mechanical) is not provided and has the potential for reversibility[1]. The physiological characteristics of critical illness have significant overlap across a wide range of presenting diseases, challenging commonly used disease-related taxonomies[2]. Interventions that target such derangements are therefore likely to impact on a wide range of diseases. Multiple diverse stressors result in a unifying state of altered tissue metabolism and bioenergetics, compounding organ dysfunction and cell death in multiple tissues such as the brain, lung, kidney and skeletal muscle[3–6].

[1]William Harvey Research Institute, Faculty of Medicine & Dentistry, Queen Mary University of London, London, UK. [2]Department of Dietetics, Adult Critical Care Unit, Royal London Hospital, London, UK. [3]Department of Nutrition and Dietetics, St Thomas' NHS Foundation Trust, London, UK. [4]Department of Critical Care, Guy's and St. Thomas' NHS, London, UK. [5]Adult Critical Care Unit, Royal London Hospital, London, UK. [6]Department of Critical Care, Bristol Royal Infirmary, Bristol, UK. [7]Genetic & Genomic Medicine Department, UCL Great Ormond Street Institute of Child Health, London, UK. [8]Developmental Biology & Cancer, UCL Great Ormond Street Institute of Child Health, London, UK. [9]Centre of Metabolism, Aging & Physiology (COMAP), MRC-Versus Arthritis Centre for Musculoskeletal Aging Research & NIHR Nottingham BRC, University of Nottingham, Nottingham, UK. [10]University College London (UCL), London, UK. [11]UCL Hospitals NHS Foundation Trust (UCLH), National Institute for Health Research (NIHR) Biomedical Research Centre (BRC), London, UK. ✉e-mail: Z.puthucheary@qmul.ac.uk

Specifically, substrate utilisation in the tri-carboxylic acid (TCA) cycle is impaired in critical illness, with tissue hypoxia and inflammation preventing glucose-derived pyruvate from being converted to acetyl-CoA, as a result of the Pasteur effect[7,8]. Amino acids may be recycled for pyruvate reconstitution in starvation, but such processes (e.g., the Cahill cycle) are affected by tissue hypoxia, inflammation, impaired Glucose Transporter Type 4 (GLUT-4) translocation, exogenous insulin therapy and other hallmarks of critical illness[8–10]. Finally, peripheral mitochondrial fatty acid oxidation is downregulated, and the resultant inability to use any of these three substrates efficiently leads to a bioenergetic crisis[8,11,12].

Hepatic metabolism of fatty acids can yield ketone bodies such as beta-hydroxybutyrate and acetoacetate. Under conditions of physiological stress, these act as substrates for Adenosine Triphosphate (ATP) generation in the mitochondria of extra-hepatic tissues. In high intensity exercise, ketogenic diets provide ketone body substrates, improving ATP production, decreasing muscle protein breakdown and improving physical performance[13]. During periods of starvation, brain metabolism relies on ketone bodies instead of fat or glucose[14], and ketone bodies may provide up to 50% of total body basal energy, enabling the high-energy requirement of the human brain to be met whilst sparing muscle[15,16]. Ketogenic diets reverse the metabolic defects of non-alcoholic fatty liver disease[17]. Patients with diabetes use ketone bodies for cardiac ATP synthesis[18].

Ketolysis occurs in mitochondria of extra-hepatic tissues, resulting in the formation of acetyl-CoA. The rate-determining step is the reconstitution of acetoacetyl-CoA from acetoacetate by the enzyme succinyl CoA-oxoacid transferase, which is not regulated by hypoxia or inflammation, unlike pyruvate dehydrogenase kinase[19]. Ketone bodies may therefore offer an alternative substrate source for energy production in critically ill patients. In addition, ketones may have other beneficial impacts in critically ill patients: in those with Acute Respiratory Distress Syndrome (ARDS), beta-hydroxybutyrate metabolically reprogrammes T-cells to improve functionality[20].

One specific physiological consequence of critical illness that may be attenuated by a ketogenic diet is that of muscle wasting. Critically ill patients lose 2–3% of their muscle mass per day[21]. This is associated with increases in length of stay and mortality, and associated physical functional disability may persist for up to 5 years[22]. Patients, carers and health services are burdened by this physical function disability, which is recognised as a public health issue[23–25]. Muscle wasting and subsequent weakness in these patients has proven resistant to all forms of exercise rehabilitation and increased nutritional delivery of energy and protein[26,27]. This loss of muscle mass appears driven by a decrease in muscle protein synthesis and unchecked muscle protein breakdown[21]. This in turn may be a consequence of bioenergetic failure and a lack of ATP production[8]. Muscle protein synthesis is a highly energy-dependent process and is likely to remain depressed until muscle bioenergetics normalise.

However, the feasibility and safety of achieving ketosis in unstable patients in multi-organ failure has yet to be proven. Ketoacidosis might occur if ketones were not metabolised, exacerbating pre-existing systemic and cellular acidosis that carries a mortality risk to critically ill patients. Ketogenic diets minimise exogenous glucose delivery, which might predispose patients to hypoglycaemia, which is harmful to patients[28]. Lastly, a ketogenic high lipid feed might increase the risk of vomiting (and therefore pulmonary aspiration), diarrhoea and pancreatitis[29].

We therefore performed a randomised trial to determine the feasibility and safety of delivering a ketogenic enteral feed in critically ill patients and collecting physical function-specific outcomes. We additionally performed an exploratory analysis of plasma metabolomic profiling to ascertain the presence or absence of a signal for efficacy in altering tissue metabolism, which might warrant further research in a larger trial.

# Results

## Safety, feasibility and tolerability

Participants were recruited between 26th September 2019 and 22nd April 2021 (including two COVID-19-pandemic related pauses) from two United Kingdom ICUs. Trial follow-up was completed by the end of April 2022. The CONSORT flow chart is available in the Supplementary Information (Fig. S1). A total of 293 patients were screened, with screening continuing until 29 patients were randomised after meeting inclusion criteria (see Supplementary Information Table S3) giving a refusal rate for assent of 4.1% (12 patients). The rate of recruitment was 2.2 participants/month for the 13 months enrolment period (Fig. S2). Participant retention rate was 82.8% (24 patients). Reasons for withdrawal are shown in Supplementary Information Table S4.

Participant demographics are shown in Tables 1 and 2. At the end of the intervention period, the number of days of vasopressor support and total daily propofol dose were higher in the control arm.

**Process and feasibility of nutritional delivery.** All patients randomised to the intervention received ketogenic enteral feeding. Feedback was obtained from 23 staff (4 research nurses, 16 ICU nurses, 1 pharmacist, 1 dietitian, 1 ICU consultant). The trial process was considered acceptable and feasible (see Supplementary Information Fig. S3), although the preparation of the modular feed was considered laborious. A mean score of 8/10 (with 10 scored as the most positive response) was obtained for the question '*How keen would you be to work on another similar study?*'

**Serious Adverse Events.** Four serious adverse events (SAEs) were reported, and all were deemed to be unrelated to the intervention. Details of these are available in the Supplementary Information (Table S5). No episodes of pulmonary aspiration were reported.

**Adverse Events.** Similar proportions of gastrointestinal events were reported between arms, with the exception of diarrhoea. The proportion of patients with diarrhoea was greater in the ketogenic enteral feeding arm (intervention vs. control 76.9% vs. 53.3%) but the difference in proportion of daily episodes less marked (53.5% vs. 42.9%). One patient in the intervention arm was transferred to total parenteral nutrition on Day 6 as a result of concerns regarding enteral feeding intolerance (Supplementary Information Table S6).

Mean base excess and bicarbonate level were similar between arms, remaining within the normal ranges (+2 to −2) and (22 mmol/L−29 mmol/L) respectively. One patient from each arm developed Acute Kidney Injury (Supplementary Information Table S7).

**Development and Establishment of Ketosis.** Ketosis was achieved within 48 hours and sustained for the 10-day intervention period, (Fig. 1A, B and Supplementary Information Fig. S4). Medium chain fatty acid (octanoic acid and decanoic acid) concentrations from the ketogenic feed were higher in the intervention arm. As expected, no differences were seen between arms in dodecanoic acid concentrations, which were not part of either feed (Supplementary Information Fig. S5). As a result, the ratio of octanoic acid to dodecanoic acid (C8:12) was higher in the intervention arm over time (Supplementary Information Fig. S5D).

**Glucose control.** Two hypoglycaemic events were reported in the control arm and none in the intervention arm (1.6% vs 0.9% respectively). Hyperglycaemia occurred in fewer patients in the ketogenic enteral feeding arm (intervention vs. control 26.9% vs. 57.5%). In keeping with this, the coefficient of variation of daily glucose was lower in the intervention arm (9.4% vs. 14.8%, Fig. 2A) as was median (IQR) cumulative insulin use (0IU (IQR 0−16) vs.78 IU (IQR 0-412). Fig. S6 shows the raw glucose data across arms.

**Table 1 | Patient characteristics and demographics**

|  | All $n = 29$ | Ketogenic feeding ($n = 14$) | Standard feeding ($n = 15$) | P value |
|---|---|---|---|---|
| Age, y | 52.0 (45.5-58.5) | 51.6 (41.8–61.5) | 52.3 (42.5–62.1) | 0.805 |
| Male, No. (%) ¥ | 17 (58.6) | 7 (50.0) | 10 (66.7) | 0.362 |
| LOS prior to ICU Admission, d# | 0.0 (0–32) | 0.25 (0–32) | 0.0 (0–22) | 0.460 |
| Ventilator days# | 11.0 (2–31) | 8.5 (4–20) | 14.0 (2–31) | 0.164 |
| ICU LOS, d# | 14.0 (2–42) | 13.0 (6–37) | 20.0 (2–42) | 0.130 |
| Hospital LOS, d# | 40.0 (2–108) | 31.5 (9–108) | 48.0 (2–106) | 0.155 |
| APACHE II | 19.1 (16.8–21.4) | 18.2 (15.5–21.0) | 21.6 (18.4–24.8) | 0.101 |
| Admission SOFA | 10.0 (9.0–11.0) | 10.1 (8.7–11.6) | 9.9 (8.4–11.4) | 0.621 |
| ICU Survival, No. (%) ¥ | 25.0 (86.2) | 12.0 (85.7) | 13.0 (86.7) | 0.941 |
| Hospital Survival, No. (%) ¥ | 21.0 (72.4) | 9.0 (64.3) | 12.0 (80.0) | 0.344 |
| RRT, No. (%) | 6.0 (20.7) | 3.0 (21.4) | 3.0 (20.0) | 0.924 |
| NMBA use, d# | 0.0 (0–9) | 0.0 (0–4) | 1.0 (0–9) | 0.203 |
| Hydrocortisone dose, mg $ # Day 1 | 0.0 (0–200) | 0.0 (0–200) | 0.0 (0–200) | 0.973 |
| Hydrocortisone dose, mg Total by day 10$ # | 0.0 (0–4000) | 0.0 (0–693) | 0.0 (0–4000) | 0.754 |
| Statin use, No. (%) | 5.0 (17.2) | 3.0 (21.4) | 2.0 (13.3) | 0.564 |
| Gastro–protection, d# | 5 (0–10) | 5.5 (1–9) | 4.0 (1–10) | 0.428 |
| Vasopressor support, d# | 4.0 (0–10) | 3.5 (0–9) | 7.0 (0–10) | 0.030 |
| Sedation use, d#s | 7.0 (1–10) | 6.5 (2–10) | 10.0 (1–10) | 0.228 |
| Total propofol dose by day 10, g | 11.1 (6.1–16.0) | 6.7 (1.4–11.9) | 15.2 (6.9–23.5) | 0.034 |

*ICU* intensive care unit, *APACHE II* Acute Physiology and Chronic Health Evaluation score, *y* year, *d* day, *No* number, *LOS* Length of Stay, *RRT* Renal Replacement Therapy, *NMBA* Neuromuscular Blockade Agent; $ Corticosteroid dosing as hydrocortisone equivalents. Data are mean (95% Confidence Intervals), except for # indicating median with range. Two-tailed Student's T-test was used except for ¥ (Chi-squared) and # (two-tailed Mann–Whitney U).

**Nutritional adequacy and substrate utilisation.** In per-protocol analysis, participants receiving control enteral feeding met 90.4% and 79.3% of energy and protein targets respectively; patients receiving ketogenic enteral feeding received 83.3% and 84.4% of energy and protein targets respectively (Fig. 2B). This was similar in the Intention-To-Treat group, although a lower proportion of control arm participants met their protein target compared with the ketogenic enteral feeding arm (71.4% vs 88.1%). Indirect calorimetry was performed on a subset of patients. RQ was 0.83 in the control patients ($n = 6$) and 0.78 in the intervention arm ($n = 8$) (Supplementary Information Fig. S7).

Plasma pyruvate concentrations were similar between arms (Fig. 2C). Plasma lactate concentrations were lower in the ketogenic enteral feeding arm at baseline and remained lower throughout the study period Fig. 2D). Collection of 24-hour urine samples to obtain total nitrogen values was not feasible in the context of heightened infection control during the pandemic.

**Data collection completeness.** The completion rate of data collection (for blood gases, biochemistry, haematology, bedside physiology, nutritional data and propofol usage) from medical records into the electronic database was 98.7% for those participants still in the study.

**Rectus femoris ultrasound.** Twenty-seven (93.1%) patients had 73 ultrasound scans performed over the period of the study. Scans were not performed in 2 (6.9%) of patients due to transfer to palliative care before the Day 1 scan was performed. However, scan quality did not reach acceptability in scans on 61 days (in 25 patients) when examined independently (DB and ZP). Reasons for this included the turnover and disruption of shift patterns of research nurses during the pandemic, and difficulties of working in protective equipment (PPE) leading to issues with consistent training and quality control.

**Physical functional outcomes.** The Chelsea Critical Care Physical Assessment Tool (CPAx) was completed in 26/29 (90%) patients at ICU discharge and 17/29 (59%) at hospital discharge. The median Chelsea Critical Care Physical Assessment score at hospital discharge was higher in the ketogenic feeding arm (34 (95%CI 22-45) vs. 25 (95%CI 8-46). Collection of data for other physical function milestones (e.g. sit-to-stand 20.8%, bed-to-chair 29.2%, Short Physical Performance Battery 9.5%, 2 and 6 minute-walk tests <5%) was severely impeded by COVID-19-related limitations in access to physiotherapists for these assessments due to re-deployment.

**Longer-term outcomes.** Quality of life assessed by EQ-5D-5L was measured at 3-, 6- and 12-months post-ICU discharge in 21 (72.4%), 21 (72.4%) and 19 (65.5%) of study participants respectively (including data from those who had died where available). Three non-responders needed a translator that was not available during the pandemic, two were in long-term care, and one had moved abroad. Primary healthcare usage data were available in 20/29 (69%) of patients. Questions of employment were abandoned due to the complexity of employment status during the COVID-19 pandemic, and the potential to cause patient distress. Final discharge location data were completed for 15/29 patients (51%).

**Metabolomic profiling**
Plasma metabolomic analyses, performed once all samples were available, were completed in February 2023. A total of 185 metabolite features were identified.

**Exploratory visualisations.** Sparse Partial Least Squares Discriminant analysis (SPLS-DA) demonstrated no difference in metabolite abundance between arms on Day 1 (Error rates>20% in all but one domain: polar positive 17%, non-polar positive 41%, non-polar negative 31%, polar negative 20%, suggesting that the plot was overfitted and not true variance;Fig. S8). By the end of the intervention period, between-arm differences were seen in 31 non-polar negative and 67 non-polar positive metabolites with Variable Importance in Projection (VIP) scores of >1. Similarly, 45 polar negative and 65 polar positive

metabolites had a VIP score >1 (all error rates <20%: polar positive 12%, non-polar positive 15%, non-polar negative 14%, polar negative 17%, suggesting plot is a result of real variation Fig. 3).

Each arm additionally demonstrated changes (VIP scores>1) in metabolite abundance over time. In the control arm, 37 non-polar negative, 39 non-polar positive, 41 polar negative and 18 polar positive metabolite abundances were differentially altered over time (Fig. S9). In the ketogenic enteral feeding arm, 23 non-polar negative, 22 non-

polar positive, 59 polar negative and 20 polar positive metabolite abundances were differentially altered over time (Fig. S10).

**Pathway analysis.** Metaboanalyst pathway analysis demonstrated differential metabolite abundance in ketogenic enteral feeding arm vs. controls in beta-alanine metabolism (Impact 0.5), glycerophospholipid metabolism (Impact 0.2) and pentose and glucoronate interconversions (Impact 0.14).

Over time, changes in metabolite abundance in the ketogenic enteral feeding arm were seen in pantothenate and CoA biosynthesis (Impact 0.1) and alpha-linoleic acid metabolism (Impact 0.33). This was different to that seen in the control arm of caffeine metabolism (Impact 0.69), terpenoid backbone synthesis (Impact 0.11) and pentose and glucoronate interconversions (Impact 0.14). These data, and non-impactful pathways are summarised in Table 3.

**Specific metabolite alterations.** Specific metabolites driving the differential pathway abundance data were then examined. When compared to the control arm, the ketogenic enteral feeding arm demonstrated increases in beta-alanine (17.2 AU (17.0–17.4) vs 16.3 AU (16.0–16.6); $p = 0.008$) and ureidopropionic acid (16.1 AU (15.7–16.4) vs 15.2 AU (14.8–15.6); $p = 0.008$) abundance and decreases in lithocholate 3-0-glucuronide (14.9 AU(14.8–15.0) vs 15.3 AU (15.2–15.4); $p = 0.004$) relative to the control feed. Differences in glycerophospholipid metabolism was driven by changes in phosphocholine residues (Fig. 4). Mean nutritional alanine delivery was similar between groups (ketogenic feed 35 g (95%CI 25–35) vs. control 27 g (95%CI 18-36); $p$–0.180).

Over time, administration of ketogenic enteral feed was associated with a difference in ureidopropionic acid abundance (16.1 AU (15.7–16.4) at day 10 vs 15.0 AU (14.7–15.2) at day 1; $p = 0.002$).In the control feed arm, differences over time in paraxanthine (15.0 AU (14.3–15.6) at day 10 vs 15.9 AU (15.3–16.5) at day 1; $p = 0.04$), palmitoyl glucoronide (16.3 AU (16.0–16.6) at day 10 vs 15.7 AU(15.5–15.9) at day 1; $p = 0.003$) and mevalonic acid (14.5 AU (13.7–15.2) at day 10 vs 15.9 AU (15.0–16.7) at day 1; $p = 0.02$) were noted.

## Discussion

We have demonstrated that inducing sustained ketosis using a ketogenic enteral feeding regimen is safe, well tolerated, and feasible in critically ill patients with multi-organ failure. Although some secondary endpoints could not be collected due to COVID-19 restrictions, recruitment and retention rates were high. Variability in glycaemic control improved, and differences between arms in terms of hypoglycaemia, insulin dosing and glucose variability all point towards a

## Table 2 | Admission diagnoses and pre-existing co-morbidities

| Admission diagnosis, No. (%) | All $n = 29$ | Ketogenic feeding ($n = 14$) | Standard feeding ($n = 15$) |
|---|---|---|---|
| Sepsis | 2 (6.9) | 1 (7.1) | 1 (6.7) |
| Cardiogenic shock | 1 (3.5) | 0 (0) | 1 (6.7) |
| Trauma | 9 (31.0) | 2 (14.3) | 7 (50.0) |
| Respiratory failure | 10 (34.5) | 4 (28.6) | 6 (40.0) |
| Intracranial haemorrhage | 12 (41.4) | 7 (50.0) | 5 (33.3) |
| Acute liver failure | 0 (0) | 0 (0) | 0 (0) |
| Acute Kidney Injury | 1 (3.5) | 1 (7.1) | 0 (0) |
| Drug overdose | 0 (0) | 0 (0) | 0 (0) |
| Emergency Surgery | 5 (17.2) | 4 (28.6) | 1 (6.7) |
| Cerebrovascular Accident | 1 (3.5) | 1 (7.1) | 0 (0) |
| Comorbidities, No. (%) | | | |
| Hypertension | 10 (34.5) | 5 (35.7) | 5 (33.3) |
| Chronic Respiratory Diseases | 6 (20.7) | 2 (14.3) | 4 (26.7) |
| Diabetes Mellitus | 6 (20.7) | 3 (21.4) | 3 (20.0) |
| Ischemic heart disease | 3 (10.4) | 2 (14.3) | 1 (6.7) |
| Psychiatric diseases | 2 (6.9) | 0 (0) | 2 (13.4) |
| Renal impairment | 1 (3.5) | 0 (0) | 1 (6.7) |
| Obesity (BMI ≥ 30 kg/m$^2$) | 4 (13.8) | 1 (7.1) | 3 (20.0) |
| Liver cirrhosis | 1 (3.5) | 1 (7.1) | 0 (0) |
| Haem-oncological disease | 4 (13.8) | 2 (14.3) | 2 (13.4) |
| Thyroid disease | 1 (3.5) | 0 (0) | 1 (6.7) |
| Crohns disease | 1 (3.5) | 1 (7.1) | 0 (0) |
| Previous Cerebrovascular Accident | 2 (6.9) | 2 (14.3) | 0 (0) |
| Chronic pancreatitis | 0 (0) | 0 (0) | 0 (0.0) |

*BMI* Body Mass Index.

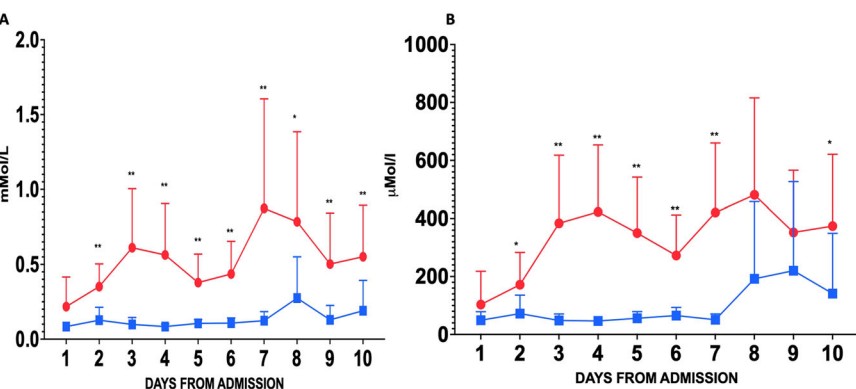

**Fig. 1 | Ketone body formation.** Plasma Beta-hydroxybutyrate (**A**) and Acetoacetate (**B**) concentrations during the 10-day intervention. Data are mean (95%CI). Red lines represent ketogenic feeding, and blue lines controls. *$p < 0.05$; **$p < 0.01$ for two-tailed Mann-Whitney-U test. $n = 14$ subjects in the ketogenic arm and $n = 15$ subjects in the control arm.

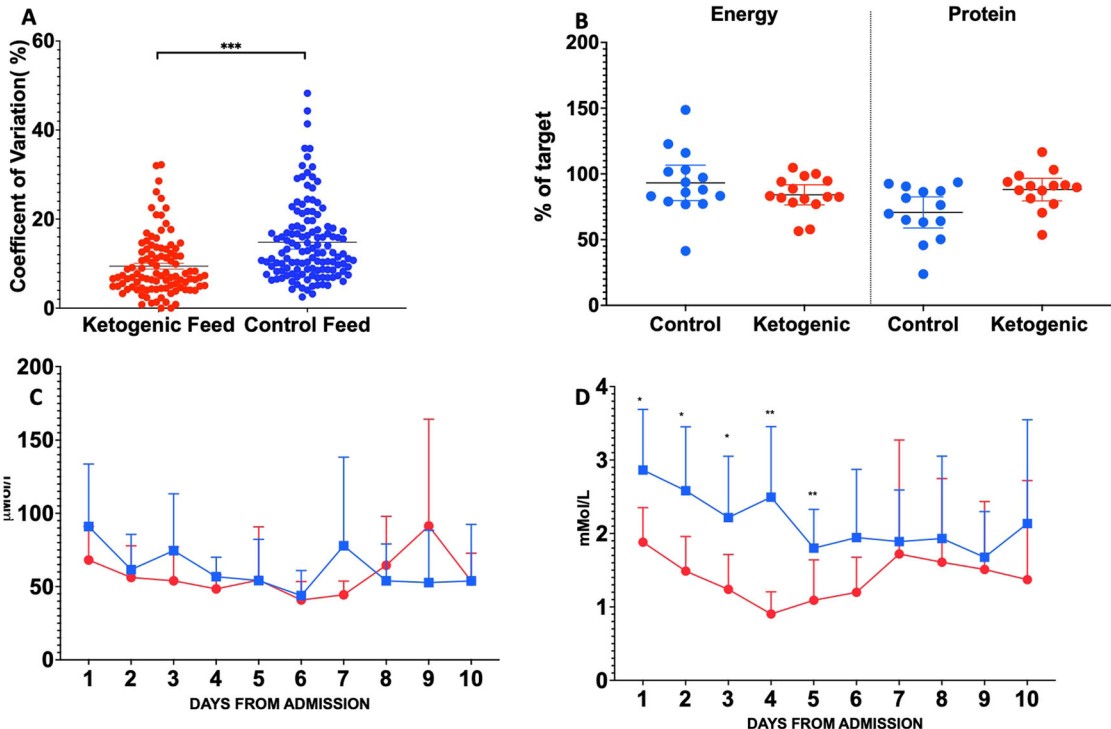

**Fig. 2 | Targetted metabolic parameters.** Co-efficient of Variation of serum glucose (**A**); Nutritional adequacy (**B**); Plasma pyruvate (**C**) and Lactate (**D**) concentrations during the intervention. Data are mean (95% CI). Red represents ketogenic feeding, blue controls. *$p < 0.05$; **$p < 0.01$; ***$p < 0.001$ between arms (two-tailed Mann–Whitney U test). $N = 14$ subjects in the ketogenic arm and $n = 15$ subjects in the control arm, (2 A 102 glucose readings in ketogenic arm vs 128 glucose readings in the control arm, $p < 0.001$).

favourable metabolic profile and ketone bodies being used as a substrate preferentially over glucose. In keeping with tissue ketone body metabolism, ketosis was not associated with development of arterial blood acidaemia. Exploratory untargeted metabolomic analyses showed a clear separation of arms at the end of the intervention. Metabolite abundance data suggest that a favourable metabolic profile developed in response to the intervention, within pathways involved in protein homeostasis and urea cycle flux. This hypothesis requires prospective testing in a larger trial.

Prior to this trial, concerns had been raised regarding the safety and feasibility of inducing ketosis in physiologically unstable patients. Ketosis is traditionally associated with pathological states in clinical medicine[30]. However, staff and patient education and engagement led to excellent rates of recruitment and retention of patients. Staff questionnaires suggest not only a high level of enthusiasm for the study, but also that this was, in the main, scalable, if a pre-made enteral feed could be sourced, of which several are available commercially.

Support for the study may be in part due to the safety profile observed, with little differences seen in adverse events or tolerability, except for the incidence of diarrhoea. We had not prespecified the definition of diarrhoea, and several such definitions exist. Diarrhoea is common in critically ill patients[29], and it is not clear whether the high medium chain triglyceride load led to an increased incidence although diarrhoea is a well-known consequence of enteral feeding when there is a high proportion of medium chain triglycerides[31]. Future trials should pre-specify such definitions for monitoring purposes. Recently a number of ketone ester supplements have become commercially available, use of which might not only circumvent the impairment of mitochondrial fat oxidation, the rate limiting process for therapeutic ketosis, and also circumvent fat intolerance. These products may or may not be feasible to use in critically ill patients and require testing.

Octanoic Acid and Decanoic Acid were delivered as part of the intervention, and the increased presence of these in the circulation supports data from recent stable isotope studies that gastrointestinal absorption is not a limiting factor[32]. Moderate levels of both medium chain fatty acids and ketone bodies in the participants receiving ketogenic enteral feeding suggest that the pathway of medium chain triglyceride lipolysis, absorption, hepatic ketogenesis and extrahepatic ketone body utilisation is functionally intact and operative in these patients, supporting the hypothesis that provision of a ketogenic enteral feed to critically ill patients provides an alternative metabolic substrate. Dodecanoic acid is not present in either the ketogenic or standard enteral feeds and its concentrations were no different between arms, acting as a form of internal control. Despite the altered composition of the enteral feed formula, nutritional adequacy was achieved equally across arms. In an exploratory analysis the respiratory quotient (RQ) was lower in the ketogenic enteral feeding arm, adding further data suggestive of ketone metabolism occurring for energy generation[33]. The equivalence in pyruvate concentrations despite reduced glycolysis may result from ketone derived acetyl-CoA production leading not only to better ATP production, but also negative feedback on pyruvate influx via pyruvate carboxylase activation[34]. Whilst the lower lactate levels see in the intervention arm is an attractive signal of substrate switching, these data are likely the result of baseline variation, and the value of lactate measurements in ketogenic feeding needs to be assessed in a larger cohort. The collection of outcome data was significantly impacted by the COVID-19 pandemic. Bedside collection of 24-hour urinary data (and hence nitrogen levels) and reliable muscle ultrasound measurements were not possible due to a combination of infection control requirements, the use of PPE, and staff shortages due to redeployment. Routinely collected physical outcome data were limited to the minimum possible as a result. Acceptability of the study protocol to staff indicated some level of

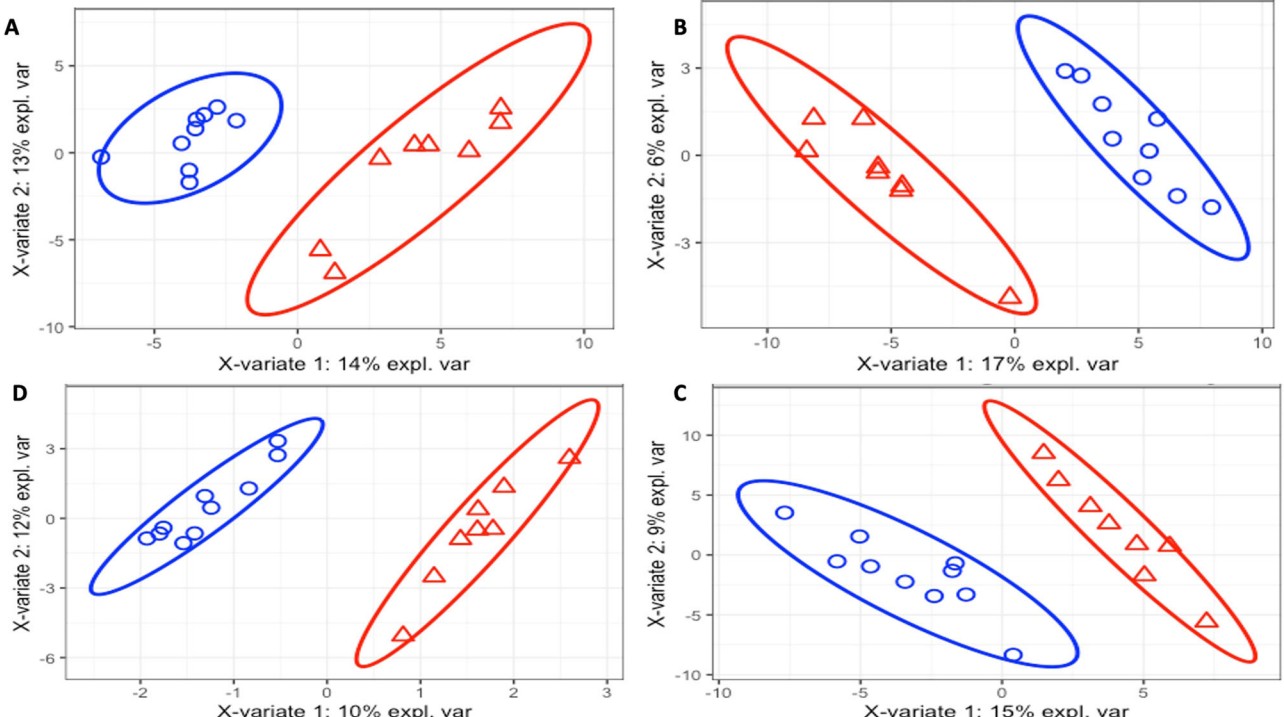

**Fig. 3 | Sparse Partial Least Squares Discriminant analysis of patient randomised to ketogenic feeding on Day 10 (red triangle) and control feeding on day 10 (blue sphere).** Clockwise from top right: **A** polar positive, **B** non-polar positive, **C** non-polar negative, **D** polar negative. Error rates are <20% (12%, 15%, 14% and 17% respectively), suggesting plot is a result of real variation. $n = 14$ subjects in the ketogenic arm and $n = 15$ subjects in the control arm.

difficulty with collecting the quality-of-life data both retrospectively (pre-ICU admission) and following ICU discharge. This is highly likely to be related to difficulties in communicating with family members due to COVID-19-related suspension of hospital visiting.

Of note, measurement of muscle mass was neither an essential nor recommended outcome measure in the Core Outcome Set for metabolic and nutritional interventions in critically ill patients and would therefore be unlikely to form part of our subsequent efficacy trial[35].

Exploratory metabophenotyping demonstrated good metabolic separation between arms following the 10-day intervention. Given the multiple tissue metabolites contributing to plasma metabolite profiles, this separation lends weight to the hypothesis that ketone bodies are being used for substrate metabolism in diverse physiological pathways. Pathways unrelated to metabolism of nutritional lipids were differentially regulated implying tissue metabolism was additionally altered. Alterations in Cahill cycle flux result in a differential metabolite abundance in the alanine pathway, suggesting a decrease in muscle protein breakdown for alanine production[10]. This is supported by alterations in Ureidopropionic acid, a urea derivative of beta-alanine and therefore a marker of said flux. Increase urea cycle flux has been previously described in critically ill patients[36,37]. Altered alanine and urea cycle flux have also been seen in healthy subjects in response to ketogenic diets though the relevance of this is unclear[38]. Alterations in the abundance of Phosphocholine residue have been noted with ketogenic diets in healthy volunteers, and are hypothesised to be related to carnitine metabolism and mitochondrial fatty acid transport, offering further support to the presence of the uptake and metabolism of decanoic (partially carnitine dependent) acid to maintain ATP production[38]. Lower phosphocholine abundances which may reflect the dysregulation of beta-oxidation have been both described and linked to mortality in critically ill patients[39–41]. Pentose and glucoronate interconversions suggest a differential regulation of hepatic

detoxification potentially related to amino acid breakdown products such as ammonia, and glucoronate abundance differences has been seen previously comparing critically ill trauma patients to healthy volunteers[42,43]. Ketogenic diets result in altered pentose and glucoronate interconversions in animal models[44]. Lithocholate Glucuronide is a bile acid metabolite, which may be related to ketogenic diet-related early satiety via Glucagon Like Peptide-1 activation, although differential bile salt abundance was not seen relative to controls in observational studies[42,45].

This study has considerable strengths. First, the very comprehensive prospective data collection gives a high level of confidence in the safety data for future trials, and suggests that such trials are unlikely to require such extensive data collection (thus reducing the data collection burden). Second, the extensive biochemical and metabolomic analysis gives insight into metabolic processes brought into play by the delivery of ketogenic enteral feeding. Untargeted metabolomic profiling was used in an exploratory fashion, to generate potential panels for future targeted work, coupled with targeted ketone and medium chain fatty acid analyses. The advantage of this approach was seen in the identification of panels separate from those that we would have used a priori e.g. energetic intermediate pathways such as TCA Cycle metabolites and NAD Metabolites. In our future trials, additional targeted panels will include those pertaining to urea cycle flux, glucoronate metabolism, carbohydrate and lipid disposal and ketone metabolites. Limitations include the need to focus on safety and feasibility which determined the sample size. Imbalances in the study cohort between arms are likely driven by this (e.g., APACHE score inequality versus SOFA equality). Regardless, all patients recruited were in multi-organ failure, and at risk of both altered substrate utilisation, muscle wasting, and subsequent physical functional impairment. All signals reported relating to efficacy should be viewed as hypothesis generating only as this trial was not powered to detect these endpoints. A further limitation relates to the

**Table 3 | Pathway Impact Scores following mapping of metabolites to relevant pathways**

| Pathway | p value | Adjusted p value | Pathway Impact |
|---|---|---|---|
| Beta-Alanine metabolism | 0.006 | 0.027 | 0.5 |
| Glycerophospholipid metabolism (RP positive) | 0.035 | 0.124 | 0.2 |
| Glycerophospholipid metabolism (RP negative) | 0.111 | 0.140 | 0.1 |
| Pentose and glucuronate interconversions | 0.161 | 0.297 | 0.14 |
| Pyrimidine metabolism | 0.020 | 0.060 | 0.01 |
| Nicotinate and nicotinamide metabolism | 0.084 | 0.161 | 0.00 |
| Glycerolipid metabolism | 0.089 | 0.161 | 0.01 |
| Propanoate metabolism | 0.126 | 0.189 | 0.00 |
| Glycine, serine and threonine metabolism | 0.176 | 0.226 | 0.00 |
| Arginine and proline metabolism | 0.201 | 0.226 | 0.01 |
| Purine metabolism | 0.321 | 0.321 | 0.00 |
| Pantothenate and CoA biosynthesis | 0.005 | 0.027 | 0.05 |
| Arachidonic acid metabolism | 0.047 | 0.238 | 0.02 |
| Linoleic acid metabolism | 0.047 | 0.238 | 0.00 |
| Sulphur metabolism | 0.075 | 0.250 | 0.00 |
| Alpha-Linoleic acid metabolism | 0.119 | 0.297 | 0.00 |
| Ether lipid metabolism | 0.178 | 0.297 | 0.00 |
| Glycerophospholipid metabolism | 0.298 | 0.399 | 0.09 |
| Pyrimidine metabolism | 0.319 | 0.399 | 0.01 |
| Purine metabolism | 0.476 | 0.528 | 0.01 |
| Taurine and hypotaurine metabolism | 0.026 | 0.089 | 0.00 |
| Glycosylphosphatidylinositol biosynthesis | 0.044 | 0.089 | 0.00 |
| Primary bile acid biosynthesis | 0.14 | 0.14 | 0.01 |
| Linoleic acid metabolism | 0.041 | 0.124 | 0.00 |
| Alpha-Linoleic acid metabolism | 0.104 | 0.168 | 0.00 |
| Glycosylphosphatidylinositol biosynthesis | 0.112 | 0.168 | 0.00 |
| Sphingolipid metabolism | 0.163 | 0.196 | 0.00 |
| Arachidonic acid metabolism | 0.264 | 0.264 | 0.00 |

Performed by over representation analysis (ORA) using MetaboAnalyst. Pathway Impact scores represent an objective estimate of the importance of a given pathway relative to the global metabolic network[59]. A cut off value of 0.1 was used, in keeping with previous work across multiple comparisons to filter less important pathways[59,60]. No other adjustments were made for multiple comparisons. p values were calculated using a one tailed hypergeometric distribution test.

missing data for some of the physical outcomes – an indication of the strain put on healthcare systems by the COVID-19 pandemic. From a feasibility perspective, future trials would require dedicated time and funding for this, as these data expose the fragility of using routinely collected physical outcome data. A lack of data on functional impacts might be considered a weakness. However, this study was primarily intended to provide feasibility and safety data, while also garnering mechanistic/physiological data upon which to base design of a trial adequately powered to detect impact on meaningful functional outcomes.

In conclusion, ketogenic enteral feeding in critically ill patients with multi-organ failure is safe and feasible. Patients who received ketogenic enteral feeding developed a different metabolic profile from controls. The efficacy of this altered metabolic profile in improving patient outcomes requires testing in prospective trials.

## Methods

We performed a single-blinded randomised controlled feasibility trial in two UK intensive care units (ICUs), with an allocation ratio of 1:1. The study conforms with the Declaration of Helsinki, and received ethics committee approval (National Research Ethics Service Committee Wales 5 – Bangor; REC reference 19/WA/0209; IRAS project ID 266031) and was publicly registered prior to the first patient being randomised (ClinicalTrials.gov, NCT04101071). We used the CONSORT (Consolidated Standards of Reporting Trials) statement when reporting this trial[46]. Patients were recruited between October 2019 and April 2021.

### Participants
Adult (≥18 years) ICU patients were screened for inclusion on weekdays, being eligible for enrolment up to 48 hours after ICU admission. Potential participants were screened by research nurses and recruited by a member of the research team.

**Inclusion Criteria.** Requiring enteral nutrition via nasogastric tube; expected to be intubated and ventilated ≥48 hours; multi-organ failure (Sequential Organ Failure Assessment (SOFA) score >2 in >2 domains)[47]; likely ICU stay ≥5 days and likely survival ≥10 days (assessed as previously by senior ICU clinicians[21]).

**Exclusion Criteria.** Primary neuromyopathy or significant neurological impairment at the time of ICU admission that would preclude physical activity; unilateral/bilateral lower limb amputation; requirement for sole or supplementary parenteral nutrition; need for specialist nutritional intervention; known inborn error of metabolism; participation in another clinical trial. Patients at risk of refeeding syndrome (based on NICE guidelines[48]) were assessed on an individual basis.

Prospective informed assent was by nominated personal consultee (in person or by telephone) or professional consultee. Retrospective participant consent was obtained on return of each participant's mental capacity. Permission to use participants' data if capacity did not return or if they did not survive, was included in the assent process.

### Feeding regimens
The ketogenic enteral feed was reconstituted for each patient in a clean kitchen area of the ICU by research nurses, with the proportions of individual nutritional components used devised by a dietitian using K.Quik® (Vitaflo International Ltd, Liverpool, UK), Renapro® (Stanningley Pharma, Nottingham, UK), Maxijul® (Nutricia, Liverpool, UK) and Fresubin® 5 kcal (Fresenius Kabi, Dublin, Ireland, if additional fat was needed). Ketogenic and standard enteral feeding regimens were provided continuously as per the standard protocol for each trust.

Patients were ineligible if they received ≥12 hours of standard feed prior to randomisation. A dietitian assessed individual patients' nutritional needs within 72 hours of randomisation. The Modified Penn State equation or a weight-based equation (e.g., 25 kcal/kg) was used to estimate energy targets. Protein targets were individualised to each patient with a range of 0.83–1.5 g/kg/d being used according to specific clinical need. Patients were considered to have received adequate nutrition if they achieved >80% of their prescribed targets. Ketogenic enteral feeding continued for the duration of the 10-day trial period as tolerated, before reverting to standard enteral feed, as per the clinician responsible for the patient's care. Patients in the control arm received the site-specific enteral feed as per Trust protocols with an agreed daily energy target to meet their nutritional needs. Multivitamins were administered daily in the ketogenic arm as micronutrients were otherwise not in the modular feed. Intravenous glucose was only to be administered for the emergency treatment of hypoglycaemia. Blood

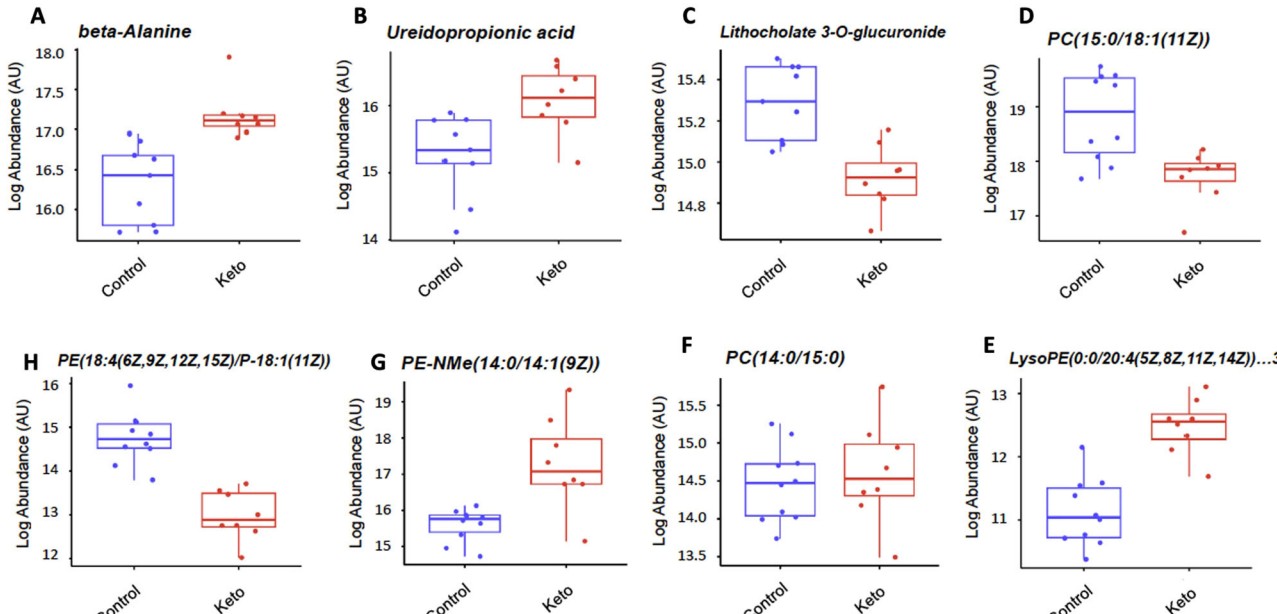

**Fig. 4 | A-H clockwise: Log abundance in Arbitrary Units (AU) of metabolites driving differential pathway analysis.** Data are mean (95%CI, Minimum-Maximum) and controls vs ketogenic: **A** Beta alanine [16 (16–17, 16–17) vs 17 (17–18, 17–18]; **B** Ureidopropionic acid [15(15–16,14–17) vs16 (16–16, 15–17)]; **C** Lithocholate 3-O-glucuronide [15(15–15, 15–16) vs 15(15–15, 15–15)]; **D** PC (15:0/18:1(11Z))[19]18–19–18–20] vs. 13(13–13,12–14)]; **E** LysoPE (0:0/20:4(5Z,8Z,11Z,14Z))[11(11–12, 10–12)vs 14(13–14, 13–14)]; **F** PC (14:0/15:0) [14(14–15, 14–15)vs15(14-15, 13–16)]; **G** PE-NMe (14:0/14:1(9Z)) [16 (15–16, 15–16) vs. 17(16–18,15–19)]; **H** PE (18:4(6Z,9Z,12Z,15Z)/p-18:1(11Z)) [15 (14–15, 14–16) vs 13(13–13, 12–14)];.

glucose levels in both feeding arms were managed according to local protocols. Given that the aim was to generate ketogenesis rather than to test a standardised feed, lipid composition was varied over time as safety and tolerance parameters allowed with medium chain triglycerides making up 40-80% of macronutrients Control participants received micronutrients as part of their standard feed, and participants in the intervention group received Sanatogen A-Z, (Bayer, UK 1 tablet daily). Further details of the feeding regimes including concentrations of medium-chain triglyceride data (Table S8) and micronutrients (Table S9) delivered can be found in the Online Supplement.

**Nutritional procedures and individualised energy and protein targets.** Nasogastric (NG) tubes were inserted as part of routine clinical care for mechanically ventilated patients. Following confirmation of correct tip position, according to national standards[48], continuous enteral feeding with either standard feed (according to local Trust protocols) or ketogenic feed was commenced.

Once reviewed by the ICU dietitian, feed volumes and compositions were adjusted according to individualised targets for energy and protein. Energy targets were mostly determined using the modified Penn State Equation (using actual body weight), the most accurate when compared to indirect calorimetry in critically ill adults[49]. Weight-based equations (e.g., 25 kcal/kg) could be used where considered more appropriate, but indirect calorimetry was not available routinely. A minimum protein target of 0.83 g/kg/day was used with adjustments for other clinical reasons (e.g., use of continuous renal replacement therapy). Ideal or adjusted body weights were used where appropriate (e.g., high body mass index) as decided by the dietitian.

**Feed management for procedures.** Where required for clinical reasons (e.g., airway management), feed was stopped according to local guidelines and restarted as soon as possible following the procedure. Where a 'feed free' time was required for enteral drug administration (e.g., phenytoin), an appropriate feeding regimen was determined by the dietitian.

**Management of high gastric residual volumes.** A Gastric Residual Volume (GRV) threshold of 300mls was set. Following administration of metoclopramide and/or erythromycin, and alleviation of any other factors which might have been reducing gastric absorption or causing ileus (e.g., drug use which could be discontinued), high GRV was managed according to local guidelines.

**Endpoints**
The primary endpoints were patient recruitment and consent rates during screening (with retention rates); the ease of reconstitution and administration of the ketogenic enteral feed by ICU and research staff (determined via questionnaire; safety (reports of adverse events [AEs] and serious adverse events [SAEs]); parameters of enteral feed absorption and blood chemistry (including glucose levels and achievement of ketosis) post-recruitment; and plasma concentrations of beta hydroxybutyrate and acetoacetate, glucose, lactate, pyruvate and medium-chain fatty acids from blood samples at timepoints during the 10-day study period (Table 4).

Secondary endpoints included arterial blood gas parameters, ultrasound-determined rectus femoris cross-sectional area (as a marker of muscle loss); non-invasive metabolic data via indirect calorimetry; and urinary concentrations of beta-hydroxybutyrate and total nitrogen, and plasma metabolomics at timepoints during the 10-day study period. Additional secondary endpoints included functional outcomes (number of days to first sit-to-stand test and to first bed-to-chair transfer prior to ICU discharge; 6-minute Walk Test and Short Physical Performance Battery at ICU/hospital discharge); clinical data (blood biochemistry; length of stay on ICU and in hospital; discharge location; number of days of mechanical ventilation; infection). Follow-up at 3-, 6-, and 12-months included Health-Related Quality of Life (EQ-5D-5L) questionnaire), employment status and primary healthcare usage costs. Given the high rate of missingness of several long-term outcomes, these will not be published here or elsewhere.

**Table 4 | Feasibility and safety outcome measures**

| Endpoint | Tool/method (reference) | Objective |
|---|---|---|
| A) Patient recruitment | From screening clinical records, medical history, demographic information; informed consent; randomisation | Feasibility of recruiting to a trial of keto-genic enteral feeding |
| B) Modular feed preparation (in ICU) to meet dietician-prescribed nutritional targets | Staff questionnaire | Feasibility of preparation of modular feed on ICUs |
| C) Administration of feed | Staff questionnaire | Feasibility of giving ketogenic feed |
| D) Glucose and lactate measurements | Point-of-care tests; routine biochemistry | Feasibility of glucose and lactose control |
| E) EQ-5D-5L questionnaire | Completed by NOK proxy either in-person or via telephone | Feasibility of determining quality of life at baseline, 3-, 6- and 12 months |
| F) Energy and protein intake | From feed recipe and volume administered | Delivery of adequate energy and protein target |
| G) Blood gases; biochemistry | Routine biochemistry analysis of blood and urine samples | Safety of ketogenic feed |
| H) Serious Adverse Events/Adverse Events | Measures of Gastric intolerance: Gastric Residual Volume; diarrhoea; vomiting; use of pro-kinetics; daily blood glucose levels | Feeding intolerance; abnormal glucose control; other Adverse Events |
| I) Muscle mass | $RF_{CSA}$ measurements | Feasibility of bedside ultrasound scans |
| J) Functional tests | Physiotherapy assessments: CPAx; 6-MWD; SPPB | Effect on physical function |
| K) Levels of ketone bodies, pyruvate, medium/long chain fatty acids and metabolites | GCMS and HPLC analysis of plasma samples | Induction of ketosis within study period; effect on nutrient metabolism |
| L) Blood urea levels in plasma | Plasma samples | Effect on protein metabolism |
| M) Indirect calorimetry | Calorimeter | Effect on basal metabolic rate |
| N) Completion of CRF | Review of medical records | Feasibility of collecting data |

*ICU* Intensive Care Unit *NOK* Next of kin, *$RF_{CSA}$* Rectus femoris muscle cross-sectional area, *CPAx* Chelsea Critical Care Physical Assessment Tool, *6-MWD* 6-Minute Walk Distance, *SPPB* Short Physical Performance, *GCMS* Gas Chromatography-Mass Spectrometry, *HPLC* High-Performance Liquid Chromatography.

## Sample size

Sample sizes of 12 per arm have been recommended where previous data on which to base a power calculation are lacking[50]. We aimed to recruit at least 37 patients to allow for a possible high drop-out rate from early death and early recovery, and for protocol violations (common in many critical care trials), and to thus leave12 patients per arm. From our previous multi-centre trials and observational studies in critically ill patients with multi-organ failure[21,51], drop out levels can be of this magnitude due to early death post-recruitment, unexpected discharge from ICU, extubation earlier than expected, protocol violations and clinical factors (such as a patient not meeting nutritional requirements) leading to the managing clinician transferring them to standard feed.

## Randomisation and blinding

Randomisation by a member of the research team took place once assent was obtained; feeding commenced as soon as possible after randomisation (according to standard clinical timing). An independent remote electronic web-based random allocation service was used to generate an unpredictable treatment group allocation and to conceal that outcome from the members of the research team until assignment occurred. Investigators performing muscle ultrasound undertook inter-observer variability assessments after training and prior to the trial commencing. All images taken during the trial were allocated an anonymised code. The off-site investigator (DB) analysing the muscle ultrasound scans was blinded to the feeding regimen allocation until post-analysis. Likewise, the majority of secondary outcomes (e.g., length of stays and days of ventilation) were collected from paper or electronic systems not susceptible to bias.

**Allocation imbalance management.** To correct an imbalance in numbers per arm, ethical approval was obtained so that randomisation could continue until 37 patients in total had been recruited.

## Criteria for premature withdrawal

In the event of the attending consultant or critical care dietitian having any clinical concerns relating to a patient in the ketogenic arm, a switch to standard feeding occurred. Feeding guidelines

included the use of prokinetic drugs, but ultimately their use was at the discretion of the treating clinicians, as is the case for routine clinical care. In the event that a protocol deviation occurred, data continued to be accrued to inform future studies. Protocol violations included meeting less than 80% of prescribed energy and protein over the period of the study protocol (10 days), cross-over between study arms, and need for parenteral nutrition or post-pyloric feeding.

## Safety and adverse event management

Where an adverse event was considered serious (SAE) or unexpected (i.e., not listed in the protocol as an expected occurrence), the Principal Investigators at both sites were mandated to report it to the Chief Investigator, who had responsibility for informing the Sponsor's research and development department within 24 hours and the main Research Ethics Committee (REC) within 15 days using the REC's standard template. All other Adverse events (AEs) were reported to Barts Health NHS Trust. Measures of adverse safety impacts included daily rates of vomiting (>10mls); number of episodes of diarrhoea (Bristol Stool Score ≥5[52] on 3 or more days) and daily rates of diarrhoea; daily rates of high gastric residual volume (GRV ≥ 300 ml), or impaired glycaemic control. Normoglycaemia was defined as a blood glucose concentration of 4–10 mmol/l, and thus concentrations of ≥10.1 or ≤3.9 mmol/l as hyperglycaemia or hypoglycaemia respectively.

**Definitions.**
- Adverse Events are any untoward medical occurrence or effect in a patient participating in the trial, which does not necessarily have a causal relationship with trial treatment. An Adverse Event can therefore be any unfavourable symptom or disease temporally associated with the use of the trial treatment, whether or not it is related to the allocated trial treatment.
- Serious Adverse Events (SAEs) are AEs that

    – Result in death;
    – Are life-threatening;

- Require in-patient hospitalisation or prolongation of existing hospitalisation;
- Result in persistent or significant disability/incapacity; or
- Result in a congenital anomaly/birth defect
- All AEs were graded for severity:

0. None: indicates no event or complication.
1. Mild: complication results in only temporary harm and does not require clinical treatment.
2. Moderate: complication requires clinical treatment but does not result in significant prolongation of hospital stay. Does not usually result in permanent harm and where this does occur the harm does not cause functional limitation to the patient.
3. Severe: complication requires clinical treatment and results in significant prolongation of hospital stay, permanent functional limitation.
4. Life-threatening: complication that may lead to death.
5. Fatal: indicates that the patient died as a direct result of the complication/adverse event.

*7.2 Potential "expected" AE's in patients with multi-organ failure*

- Abdominal distension – new, clinically significant change in appearance. Considered severe if acute obstruction.
- Abdominal pain – new, localised to abdomen and requiring more than just simple analgesia. Considered severe if not controlled with opiates.
- Electrolyte disturbance – new change that is clinically significant requiring active monitoring or treatment.
- Hypersensitivity reaction (anaphylactic reaction) – anaphylactic reaction.
- Hypoglycaemia – new, clinically significant hypoglycaemia requiring active monitoring or treatment.
- Ischaemic bowel – inferred on radiology or diagnosed visually, e.g. surgery or endoscopy.
- Nausea requiring treatment with anti-emetics; Vomiting – any episode
- Regurgitation/aspiration – any episodes

## Blood and urine analyses

Plasma concentrations medium chain fatty acids, lactate, and beta-hydroxybutyrate, whole blood acetoacetate pyruvate, and urinary beta-hydroxybutyrate were determined by gas chromatography/mass spectrometry[53]. Blood (5 mls in total) and urine (5 mls) were collected in a standardised fashion by research nurses each morning of the 10-day study period. A duplicate of each cryovial was produced.

Whole blood (2 mls) was collected into Lithium Heparin tubes. The remaining whole blood was centrifuged at 2500 g for 5 min and supernatant stored at −80 °C prior to UHPLC-MS metabolomics analysis. Blood samples (1.5 mls, in EDTA tubes) and urine samples (in universal tubes) were centrifuged at 2500 g for 5 min and the protein-free supernatant removed and stored at −80 °C, before analysis for plasma medium chain fatty acids, pyruvate, lactate and beta-hydroxybutyrate, and urinary beta-hydroxybutyrate, respectively, by gas chromatography/mass spectrometry[53].

For acetoacetate, which is not stable in plasma, 1.5 mls of whole blood was collected at the recruiting site and added to a pre-weighed plastic tube containing 5 ml of 0.77 mol/l perchloric acid previously cooled to 0 °C. After mixing, the tubes were re-weighed, centrifuged at 2500 g for 5 min and the protein-free supernatant was removed and stored. The dilution of the blood was determined by the weight changes measured. Samples were stored at -80 °C until analysis. 100ul supernatant plus 20ul internal standard mix (100uM $^{13}C_3$-pyruvate, $^{13}C_4$-acetoacetate) were added to 150ul O-(2,3,4,5,6-Pentafluorobenzyl)hydroxylamine hydrochloride (PFHBA) in 1 M HCl. After 1 hour at room temperature, 50ul concentrated $H_2SO_4$ was added to remove PFHBA, then 1 ml $H_2O$, plus 3 ml ethyl acetate. The organic

phase was transferred to a new tube, and 1 ml 0.2 N $H_2SO_4$ added to remove any residual PFHBA. The organic phase was again removed and evaporated under $N_2$. 100ul N-Methyl-N-trimethylsilyltrifluoroacetamide/1% Chlorotrimethylsilane was added, plus 50ul pyridine. After incubation at 75 °C for 1 h, samples were analysed by GC/MS (equipment as above), inlet temperature 250 °C, helium flow rate 1.5 mL/min, 2 μl injection and 1:10 split ratio. The oven temperature gradient was 100 °C, held for 1 minute, and then ramped to 190 °C at 5 °C/min, then to 300 °C at 40 °C/min. Compounds were analysed by negative chemical ionization (methane flow 2 mL/min). The following fragment ions were detected in selected ion monitoring mode: m/z 174 (pyruvate), 177 ($^{13}C_3$-pyruvate), 98 (acetoacetate), 102 ($^{13}C_4$-acetoacetate. Concentrations were corrected for blood dilution in perchloric acid.

## Ultra high-performance liquid chromatography untargeted metabolomics

Blood samples (in lithium heparin tubes) were centrifuged at 2500 g for 5 min and the protein-free supernatant removed and stored at −80 °C before transfer for metabolomic analysis. Metabolites were extracted from plasma using a dual phase Bligh-Dyer extraction and analysed using a combination of hydrophilic interaction liquid chromatography (HILIC) and Reverse Phase Ultra High-Performance Liquid Chromatography-mass spectrometry (UHPLC-MS) for polar and non-polar/lipid metabolites respectively[54].

**Liquid chromatography.** Separation was performed on an Accela UHPLC pump and autosampler. For polar metabolites, an InfinityLab Poroshell 120 HILIC-Z column (2.1 mm x 150 mm x 2.7 μm, Agilent) was used with mobile phase A 10 mM ammonium formate in 90% acetonitrile with 0.1% formic acid and mobile phase B 10 mM ammonium formate in 50% acetonitrile with 0.1% formic acid. For non-polar metabolites, a Zorbax SB-Aq RRHD column (2.1 mm x 100 mm x 1.8 μm, Agilent) was used with mobile phase A ddH₂O with 0.1% formic acid and mobile phase B methanol with 0.1% formic acid.

Table M1 Gradient conditions and flow rates for liquid chromotography

| Time | Mobile phase A (%) | Mobile phase B (%) |
|---|---|---|
| HILIC (flow rate 400 μl/min) | | |
| 0.00 | 99.00 | 1.00 |
| 1.00 | 99.00 | 1.00 |
| 3.00 | 85.00 | 15.00 |
| 6.00 | 5.00 | 95.00 |
| 10.00 | 5.00 | 95.00 |
| 10.50 | 99.00 | 1.00 |
| 15.50 | 99.00 | 1.00 |
| Reverse phase (flow rate 400 μl/min) | | |
| 0.00 | 99.00 | 1.00 |
| 0.50 | 99.00 | 1.00 |
| 2.00 | 50.00 | 50.00 |
| 10.00 | 1.00 | 99.00 |
| 12.00 | 1.00 | 99.00 |
| 12.50 | 99.00 | 1.00 |
| 17.50 | 99.00 | 1.00 |

*HILIC* Hydrophilic interaction chromatography.

**Mass spectrometry.** Data acquisition was performed by a Q-Exactive high-resolution mass spectrometer (Thermo Fisher Scientific) operated in the positive and negative ionisation modes.

Table M2 Mass Spectrometry parameters for positive and negative ionisation modes

| Parameter | HILIC | | Reverse phase | |
|---|---|---|---|---|
| Mode | Positive | Negative | Positive | Negative |
| MS scan parameters | | | | |
| Scan type | Full MS | | Full MS | |
| Scan range | 70–1050 m/z | | 150–2000 m/z | |
| Fragmentation | None | | None | |
| Resolution | 70,000 | | 70,000 | |
| AGC target | 1e6 | | 3e6 | |
| Maximum IT | 100 ms | 200 ms | 100 ms | 250 ms |
| Microscans | 5 | | 1 | |
| Sheath gas flow | 54 | | 48 | |
| Aux gas flow | 13 | | 11 | |
| Sweep gas flow | 0 | 3 | 5 | |
| Spray voltage | 4 kV | 3.5 kV | 3 kV | 3.5 kV |
| Capillary temperature | 280 °C | 320 °C | 300 °C | 320 °C |
| S-lens RF level | 50 | 60 | 60 | |
| Aux gas flow heater temperature | 430 °C | 320 °C | 300 °C | |
| MS/MS scan parameters | | | | |
| Resolution | 17,500 | | 17,500 | |
| AGC target | 1e6 | | 1e6 | |
| Maximum IT | 50 ms | | 50 ms | |
| topN peaks | 3 | | 3 | |
| Isolation window | 1.5 m/z | | 1.5 m/z | |
| Normalised collision energy | 25, 60, 100% | | 25, 60, 100% | |

*HILIC* Hydrophilic interaction chromatography.

**Sample pre-processing.** Chromatograms were extracted using the centWave algorithm at a tolerance of 20 parts per million (ppm). Chromatograms were aligned using the Obiwarp method with bin size 0.6. Bandwidth was determined manually for correspondence. Missing values were imputed via integration of peak area. For each ion mode and polarity, data were exported as a data frame of metabolite feature *vs* sample ID with associated chromatographic peak areas for each detected metabolite. Metabolite features were extracted from raw files using XCMS, then quality-control filtered and normalised according to standard procedures[55,56] Validated metabolite features were annotated to provide putative metabolite ID's. Metabolites were putatively identified by metID which utilises m/z and MS² spectra matching from public metabolomics databases[57]. Mapping of metabolites to relevant pathways was performed by over representation analysis (ORA) using MetaboAnalyst 5.0 (www.metaboanalyst.ca/MetaboAnalyst/home.xhtml)[58]. Pathway Impact scores represent an objective estimate of the importance of a given pathway relative to the global metabolic network[59]. A cut off value of 0.1 was used, in keeping with previous work across multiple comparisons to filter less important pathways[59,60]

Following recommended guidelines[56], metabolite features were retained when: peaks were present in at least 70% of pooled QC samples, relative standard deviation was less than 30%, and the extraction blank to mean QC peak area was less than 50%[8]. For PLS-DA, all variables are included in the analysis, this being the primary method to reduce variable numbers using the VIP scores. Probabilistic quotient normalisation was applied to the remaining features, then missing values were imputed using the k-Nearest Numbers algorithm. Data were log transformed prior to analysis. Autoscaling was performed, where each variable was mean centred and divided by the standard deviation.

Missing data rates were as follows:

| Day 1 | Day 10 |
|---|---|
| HILIC positive - 3.07% | HILIC positive - 3.62% |
| HILIC negative - 9.19% | HILIC negative - 12.39% |
| RP positive - 2.72% | RP positive - 2.90% |
| RP negative - 4.28% | RP negative - 2.58% |

### Statistical analyses

Descriptive analysis was performed for the continuous outcomes using mean (95% confidence intervals) and Student's T-test for parametric data, or median (range) analysed using Mann Whitney U test for non-parametric data. Chi-squared testing was used for proportional data. Recruitment rates are shown as a percentage with 95% confidence interval. All analyses were performed on an intention-to-treat basis using GraphPad 8.0 (www.graphpad.com) Two-tailed tests were used, and statistical significance was indicated by $p \leq 0.05$.

For metabolomic analyses, data were centered and scaled to perform sparse partial least squares discriminant analysis (SPLS-DA) validated by k-fold cross-validation to establish differences between ketogenic and control diet groups at baseline and on day 10. Owing to the high variability in this data set, orthogonal projection to latent structures (OPLS) was utilised to maximise variation[61]. Models with error rates greater than 20% were considered to be overfitted (i.e., the model described random error in the data rather than relationships between variables)[62,63]. In non-overfitted models, all variables with a Variable Importance Projection (VIP) score greater than 1 were retained for further analysis[64]. The VIP score is a quantitative assessment of the discriminatory power of each individual feature[65].

For time course analysis, a linear mixed effect model was fitted to each data matrix using the limma package and R version 4.0.3 (https://cran.r-project.org/) and R Studio (version 1.3.1093)[66]. Diet and time were fixed effects. Participant ID was a random effect to account for subject-specific variation. Contrast matrices were set up comparing metabolite abundance at baseline and days 3, 5 and 10 of the intervention. Empirical Bayes moderated t-tests were performed to obtain *p*-values. False discovery rate (FDR) was accounted for using the Benjamini-Hochberg procedure. Metabolites were deemed significantly different between comparisons when FDR < 0.05.

### Ultrasound assessment of muscle mass

**Trial personnel training.** All trial personnel involved in image acquisition were trained onsite by a researcher experienced in the method (AM or ZAP). Personnel then underwent a period of practice and were expected to confirm inter- and intra-rater reliability in 10 healthy subjects at their local sites. Sites were deemed trained and ready to begin recruitment if images analysed independently (ZAP) were found to have an Intraclass Correlation Coefficient >0.9. However, during the pandemic training was not performed in protective equipment, and new staff did not have the same level of training since access to healthy controls was limited, as was time to train and determine the Intraclass Correlation Coefficient.

**Machines used.** Measurements were made using the following ultrasound machines: Sonosite M-Turbo, C60XI 5-20 MHz probe (FijiFilm SonoSite Ltd, London, UK) [Royal London Hospital]; Sonosite M-Turbo machine HFL 50x/15-6 MHz transducer [Bristol Royal Infirmary].

**Image acquisition.** The method for measurement was as described previously[4,67,68]. Three images were captured at each time point.

**Image analysis.** Images were stored on encrypted memory sticks under a pseudo-anonymized patient number and transferred to password-protected computers. Images were then analysed offline by a single experienced member of the research team (DB) not involved

with the clinical management of any study patients and blinded to the intervention they received (Image J software, National Institutes of Health, US). Rectus femoris muscle cross sectional area was taken as the average of three consecutive measurements within 10% of one another. Scans from 1 in 4 patients underwent re-analysis to ensure good intra-rater agreement.

## Data collection

Routinely measured physiological data were collated through the 10-day study period. Medications administered to the patient each day were recorded, as were details of energy and protein prescribed and received (from feed, propofol and additional glucose), enabling total calories and grams of protein delivered, the percentage of target calories and protein delivered, and nutritional delivery to be compared between the two arms. Where possible functional milestones (days before Bed-to-Chair transfer and Sit-to-Stand times; 6-Minute Walk Distance and Short Physical Performance Battery) were collected at ICU/hospital discharge. Length of mechanical ventilation, ICU and hospital stays, and hospital discharge destination were collected and noted from medical records. Follow-up data (health-related quality of life; employment and primary care usage) were collected at 3-,6- and 12-months post-ICU discharge if relevant staff were available given COVID-19-related limitations. Data were collected initially into a paper Case Report Form and then transferred in anonymised form to a secure database.

## Reporting summary

Further information on research design is available in the Nature Portfolio Reporting Summary linked to this article.

## Data availability

The raw data are protected and are not available due to data privacy laws. These data are not publicly available since this was not included in research participant consent. Raw deidentified data will be available post-publication for up to 2 years for non-commercial research purposes with appropriate ethical approvals for vulnerable patients by contacting the corresponding author). The processed data that support the findings of this study are available for research purposes in the Supplementary Information/Source Data file. Source data are provided with this paper.

## Code availability

Code used in this manuscript are available at: https://github.com/danwilco/Coding_Projects/tree/main/ASICS.

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

## Acknowledgements

AM, DB, SE, KR, HM and ZP received a Research for Patient Benefit grant (PB-2006) from the National Institute for Health Research (NIHR). HM received funding from the NIHR's Biomedical Research Centre (BRC) at University College London Hospitals, London, UK. The research was also supported by the NIHR BRC based at Guy's and St Thomas' NHS Foundation Trust and King's College London and by the NIHR BRC at Great Ormond Street Hospital. The views expressed are those of the authors and not necessarily those of the NHS, the NIHR, the Department of Health or the funders. Metabolomic data was supported by an educational grant from Nestle Health Sciences. Vitaflo International Ltd were involved in initial discussions about the study and provided the K.Quik® component for the ketogenic feed gratis. Neither Vitaflo International Ltd nor Nestle Health Sciences

contributed to study design, study implementation, data analysis or interpretation. We would like to thank the patients (and their families) who took part, Phil Hopkins and the Trial Steering Committee (Nicholas Hart, Ella Terblanche and Mark McPhail) and Data Monitoring Committees (Ben Shelley, David Griffiths, Jackie Cooper and Brijesh Patel), and the research nurses of both recruiting centres for their willingness to engage. Specifically: Maria Fernandez, Filipa Santos, Amaia Garcia, Fatima Seidu, Katie Sweet. We would also like to thank those who funded this study: NIHR Research for Patient Benefit (PB-PG-0317-20006: £249,560 AM, ZP; plus £10,549 additional COVID-related funding), Nestle Health Science Education grant (£25000 ZP), and Baxter Healthcare Ltd (loan of Indirect Calorimeter).

## Author contributions

A.M., D.B., S.E., K.R., H.M. and Z.P. conceived and designed the clinical trial. AM, DB,AL, KR, ZP, JP, RP, AP, TM, FS,FS, KL carried out and delivered the clinical trial. AM., ZP., PA., DW., HC., IA., SE., TB., SH. performed the analyses. All authors read and approved the manuscript.

## Competing interests

DEB has received speaker fees, conference attendance support or advisory board fees from Baxter, Cardinal Health and Avanos. ZP has received honoraria for consultancy from GlaxoSmithKline, Lyric Pharmaceuticals, Faraday Pharmaceuticals and Fresenius-Kabi, educational support from Baxter and Nestle Health Science and speaker fees from Orion, Baxter, Sedana, Fresenius-Kabi and Nestle. HM holds patents relating to intravenous hydration and to regulation of metabolic efficiency using renin-angiotensin system antagonists. SE and SJH hold patents with Vitaflo International Ltd for compositions different from that used in this study, for treating/ dietary management of drug resistant epilepsy and disorders associated with mitochondrial dysfunction, and also are in receipt of grant funding from Vitaflo International Ltd (not connected with this study). AL has received honorarium from Baxter for speaker fees. A patent has been submitted for the ketogenic feed regime used in this study (ZAP, AM, AL, DB). Vitaflo International Ltd were involved in initial discussions about the study and provided the K.Quik® component for the ketogenic feed gratis. Neither Vitaflo International Ltd nor Nestle Health Sciences contributed to study design, study implementation, data analysis or interpretation. Other authors declare no competing interests.
