## [Peer Review File · Nature Communications]

nature portfolio

Peer Review File

A pilot study of alternative substrates in the critically ill subject
using a ketogenic feedREVIEWER COMMENTS

Reviewer #1 (Remarks to the Author):

Review

Alternative Substrates in the Critically Ill Subject (ASICS): A Randomised Pilot Study to Assess the Safety, Feasibility, Tolerability and Metabolic Profiling of a Novel Ketogenic Feed
McNelly et al.

- Dear authors, thank you for the submission of your article regarding a novel ketogenic feed for critically ill patients. I found the research to be well designed and executed, particularly given the impact of Covid during data collection. The research idea is novel and some of the data are noteworthy for the field and wider audience interested in metabolism. It is noteworthy that patients given the ketogenic feed had better metabolic control and some indication of altered metabolism. The tolerability of the feed in this patient group is also good proof of concept for larger trials.

The lack of physical functional outcomes and longer-term outcomes is a limitation of the work, since the alterations in metabolism are perhaps only insightful if there is a positive change in functional outcomes and patient health. Below are some points that I believe are to be addressed to improve the manuscript. I have included an annotated manuscript for ease of understanding too.

Abstract

- Clarify what biological material was used for metabolomics in the abstract.

Introduction

- Generally, well written and addresses the problem to be addressed. There are some small type errors highlighted in the annotated file.

Methods

- Comprehensive and well written. In the feeding regimens section, please refer the reader to the supplementary material. I wondered how the protein and energy targets and so on where derived. Eventually I found the info in supplementary material. The information in the supp material is thorough so please encourage the reader to read it.

- Regarding the section 'Blood and urine analysis' - When were samples collected? Was this

performed in a consistent and controlled manner? Are these the same samples that were then used for UHPLC-MS metabolomics? Sample collection can have a large effect on the metabolome given the dynamic and unstable nature of plasma metabolites. As much information as possible is required to assess whether sampling may have impacted your results.

- Regarding statistical analyses – can you clarify what normalisation and scaling approaches were used?

- Regarding data availability statement - Presumably the metabolite data have been deposited in a repository such as <https://www.ebi.ac.uk/metabolights/>

If not, whether the metabolomics analyses planned in the research design? If the analyses were included, what is the reason not gain consent to make the data publicly available?

Results

- Regarding glucose control, did the authors calculate area under the curve? Might this be an important added insight than just reporting daily variance?

- Is it worth even reporting the RF ultrasound scan since the data can't be used?

- Are the data from the CPAX available and do they show any useful information regarding patient progress?

- Regarding specific metabolite alterations, these metabolites aren't discussed further in the discussion. Could the authors speculate as to the relevance of these key metabolites in driving the changes in pathways.

Discussion

- In the first paragraph the authors state that 'metabolite abundance data suggest a favourable metabolic profile developed in response to the intervention'. To say this, we need to know what a favourable profile looks like. Can you report what a favourable profile would be? What do healthy controls look like? Are there any published data you could use to support your assertions here?

- The authors describe the RQ data as exploratory. Why is it considered so? If the authors hypothesise that the ketones can be effectively absorbed and utilised offering an alternative fuel source in the critically ill, you could also hypothesise that RQ would be lower in ketone supplemented patients.

- Regarding the discussion of lactate, the control group also showed a decline over time and whilst absolute values differ, relative to baseline it seems like there isn't much difference

between groups. I agree that there does seem to be a switch in substrate use but not sure the lactate data support this claim. Could you expand on this and perhaps plot the data relative to the first timepoint for each treatment arm?

- Regarding the paragraph on exploratory metabophenotyping - I find this quite weak and it doesn't support the statement made in paragraph 1 of the discussion that 'a favourable metabolic profile developed in response to the intervention'. Are you able to make any comparisons with other studies that report metabophenotyping in critically ill vs healthy controls or indeed just healthy controls? Whilst you report some changes in metabolites that indicate a metabolic shift, can you confidently say that it is towards a 'healthy' metabolite profile.

Figure S7 appears to show that the two treatment arms had a fair degree of separation at baseline although the model is perhaps overfitted. Can the authors comment on whether the degree of separation observed at day 10 is due to the treatment received or perhaps partly explained by baseline separation between groups?

Whilst the sample is small, the data could be described further. Given VIPs are presented in figure 4 it would be useful to discuss their involvement in important bioenergetic pathways in some more detail. Perhaps comparison to other ketone supplementation studies that have explored pathway alterations? These exploratory type analyses need to give the reader more information on which to build new hypotheses and as it stands, I don't think this section does that very well.

When writing sentences such as 'Pathways unrelated to metabolism of nutritional lipids were differentially regulated implying tissue metabolism was additionally altered' please tell us what pathways you're referring to.

I would also suggest commenting on the utility of untargeted metabolomics in such study designs as yours. The metabolite profiles that you have generated are interesting particularly given the cohort from which they're derived.

Reviewer #2 (Remarks to the Author):

This study implemented a ketogenic diet to critically ill patients with multiorgan failure and demonstrated that this metabolic-based therapy was safe, feasible, tolerable and resulted in potentially favorable metabolomic changes.

The use of ketogenic diets for critically ill is original and represents a potentially novel indication that could advance the science and application of ketogenic metabolic therapy. It is commendable that the authors could implement this therapy in such a compromised patient population.

The study, as reported, has limitations and requires clarification on several things to be of significance to the field, related fields, and for ensuring this work could be sufficiently replicated with a standardized protocol.

The main critiques of this work are listed below:

1. The most important methodological information missing that is required to critically review this manuscript is a detailed description of the "Novel Ketogenic Feed". As with any clinical intervention, it needs to be standardized to make sense of the outcome data. A ketogenic diet is unique in that it is the only diet defined by an objective biomarker (e.g., urine, blood or breath ketones); however, there needs to be a standardized macronutrient, micronutrient, fatty acid composition of both standard and ketogenic diet formulations to understand these outcomes, especially the metabolomics. Ideally this should be in the main body of the manuscript but could also be listed as a supplement. Much like studying the effects of a drug, the intervention needs to be clearly defined and standardized for all patients (or within defined parameters). For example, the % of macronutrients (4:1, 3:1, 2:1, MAD?), composition of fats in the dietary formulation (MCT, LCT, omega-3, etc.) will have a profound effect on the metabolomics. It is essential to know the exact lipid composition of the dietary intervention. and it appears there may have been variations in patient, but it is impossible to tell. A table listing this information should be in the main body of the manuscript and is standard practice for ketogenic diet studies and listed in medical ketogenic diet textbooks (see Kossoff et. al.)
2. Regarding the secondary endpoints and blood draws, it would be important to know when these samples were collected. Was blood drawn morning, evening, after an overnight fast, after a meal? The timing of the blood draw may influence the metabolomic data.
3. The patients were reported to have multiorgan failure. Was this confirmed with bloodwork (Liver enzymes, CMP?). Ketogenic diet therapy requires the production of ketones, which is associated with robust hepatic ketogenesis. Reduced hepatic beta oxidation of fats can be a major limitation with this therapy, so it would be important to

know this information, and if it influences outcomes.

4. Line #104: Authors state that in the context of critical illness and organ dysfunction that....."mitochondrial fatty acid oxidation is downregulated, and the resultant inability to use any of these three substrates (glucose, amino acids, fats) efficiently leads to a bioenergetic crisis." This was a rationale for using the ketogenic diet, but this statement contradicts the high fat ketogenic diet approach. The production of ketones is directly dependent upon mitochondrial fat oxidation rates. If the goal is to rapidly induce and sustain therapeutic ketosis as a means to supply alternative energy (BHB and AcAc), why not directly administer dietary ketone bodies or ketone supplements? This should be mentioned in the discussion. Supplemental ketones would circumvent the impairment of mitochondrial fat oxidation, the rate limiting process for therapeutic ketosis and also circumvent the fat intolerance. Dozens of clinical trials are currently using dietary ketones to circumvent the fat intolerance that is a feature of many chronic illnesses (see clinicaltrials.gov).

Reviewer #3 (Remarks to the Author):

The authors report on the feasibility of using ketogenic enteral feeding in early critical illness. The paper is well-written with an appropriate study design used to answer the questions. Here are some minor comments.

In Table 2, some variables are summarized by allocation, using means, while others are summarized using medians. Is there any reason for this? Perhaps it would be worth stating in the methods (Statistical Analyses) that parametric or non-parametric approaches were used to compare means/medians after first checking the distribution of the data? It would be good to mention the variables that were included in the discriminant analysis, and whether there were any issues with missing data and how this was handled.

Under Process and Feasibility of Nutritional Delivery, a snapshot of the questionnaire administered to staff is given in Figure S3. It is not clear whether these are the responses from one of the participants to some of the questions in the questionnaire or not. Giving mean scores to the individual questions as was done for the question 'How keen would you be to work on another similar study?' would be more informative.

Under adverse events on page 9, the "The proportion of patients with diarrhoea was greater in the ketogenic enteral feeding arm (intervention vs. control 76.9% vs. 52.3%)" should read "The proportion of patients with diarrhoea was greater in the ketogenic enteral feeding arm (intervention vs. control 76.9% vs. 53.3%)" as 53.3% is the number in Table S6. Please indicate the N for each group for base excess and bicarbonate in S7.

Other comments:

- Naming of the figures could be simpler, e.g., Figure 2 instead of Figure2ACBD, since the sub-titles A to D can be seen on top of the individual figures.
- The labels for Figure 4 are not very clear and could be made bigger in ggplot2.
- Figure 1AB presents mean (95% CI) of Plasma Beta-hydroxybutyrate and Acetoacetate concentrations during the 10-day intervention. Similar figures are used for other measures (Figure 2, S4 and S5). Because of the small numbers or large standard errors, the lower limit of the CI cannot be shown in these figures. An alternative would be to use jittered boxplots although this might be hard to fit on the graph. The authors are correct to report p-values based on the Mann Whitney-U test in these figures.

Reviewer #4 (Remarks to the Author):

Thank you for the opportunity to review the manuscript:

Alternative Substrates in the Critically Ill Subject (ASICS): A Randomised Pilot Study to Assess the Safety, Feasibility, Tolerability and Metabolic Profiling of a Novel Ketogenic Feed

By the authors:

Angela McNelly¹, Anne Langan², Danielle E. Bear^{3,4}, Alexandria Page⁵, Tim Martin⁵, Fatima Seidu⁵, Filipa Santos⁵, Kieron Rooney⁶, Kaifeng Liang¹, Simon J Heales⁷, Tomas Baldwin⁸, Isabelle Alldritt⁹, Hannah Crossland⁹, Philip J. Atherton⁹, Daniel Wilkinson⁹, Hugh Montgomery^{10,11}, John Prowle^{1,5}, Rupert Pearse^{1,5}, Simon Eaton⁸ and Zudin A. Puthuchery^{1,5*}

The authors have to be commended for conducting this complex small RCT in challenging times during the COVID-19 pandemic.

In summary, the impaired utilization of glucose and fatty acids leading to bioenergetic failure contributes to organ dysfunction in critically ill patients. Ketone bodies have been proposed as an alternative energy source, but it is unclear if inducing a ketogenic state is safe for physiologically unstable patients. A pilot trial was conducted on 29 mechanically ventilated adults with multi-organ failure to compare ketogenic feeding with standard enteral feeding. The results showed that ketogenic feeding was feasible, safe, and well tolerated, resulting in ketosis. Patients on the ketogenic diet experienced fewer hypoglycemic events, required less insulin, but had slightly more episodes of diarrhea. Metabophenotyping analysis revealed altered Cahill cycle flux and bioenergetic states, suggesting a beneficial metabolic profile. These findings indicate that ketogenic feeding may be a viable intervention for addressing bioenergetic failure in critically ill patients.

I would like to request the authors to address the following comments:

Abstract:

Patients receiving ketogenic feeding had fewer hypoglycaemic events (0% vs. 1.58%), required less exogenous insulin (0.0 IU (IQR 0-16) vs. 78 IU (IQR 0-412) but had slightly more daily episodes of diarrhoea (53.5% vs. 42.9%) over the trial period.

1. Why is the 1,58% so detailed while the ketogenic diet is reported a 0% and not as 0,00%? Please provide consistent reporting.
2. Same for 0.0 versus 78. Please provide consistent reporting.
3. It is unclear whether differences are statistically different or only show trends?

Introduction:

The physiological characteristics of critical illness have significant overlap across a wide range of presenting diseases, challenging commonly used disease-related taxonomies

4. What is the purpose of the part of the sentence: challenging commonly used disease-related taxonomies? Is this relevant to this paper? Please provide clarification.

In high intensity exercise, ketogenic diets provide ketone bodies for substrates, improving ATP production decreasing muscle protein breakdown and improving physical performance.¹³

5. Please add a comma after "ATP production."

Introduction:

Ketone bodies may therefore offer an alternative substrate source for energy production in critically ill patients. In addition, ketones..

6. Separate the sentences with a space for clarity.

Exclusion criteria:

Patients at risk of refeeding syndrome (based on NICE guidelines³¹) were assessed on an individual basis.

7. In several observational studies among ICU patients, refeeding syndrome (RFS) is typically not identified by NICE guidelines but by phosphate drops. Could you explain why NICE criteria were used in a critically ill population?

8. Why was diabetes mellitus (DM) not listed as an exclusion criterion? Please provide an explanation.

Targets:

Protein targets were individualised to each patient with a range of 0.83 -1.5g/kg/d being used according to specific clinical need.

9. Why were the protein targets set at low levels (0.83 - 1.5g/kg/d)? ESPEN recommends a higher target of > 1.3 g/kg/day. What were the specific conditions for using lower targets in this study?

Patients in the control arm received the site-specific enteral feed as per Trust protocols with an agreed daily energy target to meet their nutritional needs.

10. It appears that the interventions were not isocaloric or isonitrogenous. Please comment on the potential impact of this on the study outcomes.

Interventions:

Multivitamins were administered daily in the ketogenic arm as micronutrients were otherwise not in the modular feed.

11. It seems evident that the ketogenic diet patients received more vitamins than the control patients. Could you provide details on the products used and their dosages? This information is essential as it could introduce a confounding factor affecting the analysis of the ketogenic diet's effect.

Monitoring:

daily rates of high gastric residual volume (GRV>300ml),

12. Why were such low gastric residual volume (GRV) thresholds used (GRV > 300ml)? Typically, abandoning the measurement or setting the threshold at GRV < 500ml is recommended. Please explain the rationale for using a lower threshold.

Sample size calculation:

Sample sizes of 12 per arm have been recommended where previous data on which to base a power calculation are lacking³⁴. We aimed to recruit at least 37 patients to allow for a possible high drop-out rate from early death and early recovery, and for protocol violations (common in many critical care trials), and to thus leave 12 patients per arm.

13. What was the basis for considering a dropout rate of 13 out of 37 patients? This dropout

rate seems very high. Was the inclusion continued until 2x12 eligible and evaluable patients were recruited? Please provide further clarification on the rationale behind the sample size calculation.

Severity of illness:

mean (95%CI) APACHE II score was higher in the control arm than the ketogenic feeding arm (21.6 (18.4-24.8) vs. 16.4 (13.5-19.3); $p=0.025$), admission SOFA scores were similar (9.9 (95%CI 8.4-11.4) vs. 10.1 (8.7-11.6); $p=0.621$).

14. It is uncommon to observe an average APACHE II score of 16.4 in severely ill ICU patients with an average SOFA score of 10.1. Could you please explain these data further and address any potential discrepancies? Are you sure these data are correct? In most studies patients with SOFA scores of 10 and higher have APACHE-II scores > 22.

Missing study parameters:

Collection of 24-hour urine samples to obtain total nitrogen values was not feasible in the context of heightened infection control during the pandemic.

15. What is the reason for not being able to collect urine during the pandemic? Could you explain the reason why collecting 24-hour urine samples to obtain total nitrogen values was not feasible during the pandemic? Was there any consideration given to collecting smaller samples (from a large volume saved in ICU) or using alternative methods? Please provide more information.

Discussion

Variability in glycaemic control improved, and differences between arms in terms of hypoglycaemia, insulin dosing and glucose variability

16. Typos: hypoglycaemia, should be hypoglycaemia.

While the significance of the equivalence in pyruvate concentrations is unclear, the lower lactate levels seen in the intervention arm are also suggestive of substrate switching.

17. Could it be that the higher acetyl-CoA resulting from the ketogenic diet conversion not only leads to better ATP production but also blocks pyruvate influx, resulting in similar pyruvate levels despite reduced glycolysis? Please comment on this potential mechanism.

Alterations in Cahill cycle flux result in a differential metabolite abundance in the alanine pathway, suggesting a decrease in muscle protein breakdown for alanine production¹⁰.

18. Was the dietary intake of alanine different between the groups? Please provide additional information and clarification.

Table 1:

19. Why is the population in this study 10-12 years younger than typically seen in European ICU studies? This seems unexpected based on the inclusion and exclusion criteria.

20. Could the effect and feasibility of the intervention be different in a more elderly population? Please discuss the limitations of external validity for the study.

21. For the variables listed in the table, could you provide the corresponding p-values or indicate which variables were statistically significant and which were not? This information would enhance the interpretation of the results.

Thank you for the opportunity to review this interesting paper.

ASICS REPLY TO REVIEWERS

REVIEWER

COMMENTS

1. Dear authors, thank you for the submission of your article regarding a novel ketogenic feed for critically ill patients. I found the research to be well designed and executed, particularly given the impact of Covid during data collection. The research idea is novel and some of the data are noteworthy for the field and wider audience interested in metabolism. It is noteworthy that patients given the ketogenic feed had better metabolic control and some indication of altered metabolism. The tolerability of the feed in this patient group is also good proof of concept for larger trials.

Many thanks for these kind and positive comments.

2. The lack of physical functional outcomes and longer-term outcomes is a limitation of the work, since the alterations in metabolism are perhaps only insightful if there is a positive change in functional outcomes and patient health.

We agree that this is a limitation of this work, which was primarily intended (as stated in the manuscript) to provide feasibility and safety data, while also garnering mechanistic/physiological data. Funding for a trial of impact on functional outcomes could not be sought before such data were obtained. With this obtained, we are now seeking funding for such a trial. Adequately powered to provide meaningful data (to detect a change of 2 repetitions in a 30-second Sit-to-Stand test, allowing for mortality, drop out etc), 276 patients will need to be recruited. We have added the following to the limitations section:

“A lack of data on functional impacts might be considered a weakness. However, this study was primarily intended to provide feasibility and safety data, while also garnering mechanistic/physiological data upon which to base design of a trial adequately powered to detect impact on meaningful functional outcomes”.

In response to later comments we have added CPAX score data to the manuscript:

“The median Chelsea Critical Care Physical Assessment score at hospital discharge was higher in the ketogenic feeding arm (34 (95%CI 22-45) vs. 25 (95%CI 8-46).”

3. Clarify what biological material was used for metabolomics in the abstract.

This has been clarified with the addition of “*plasma*”

4. Generally, well written and addresses the problem to be addressed. There are some small type errors highlighted in the annotated file.

These have now been addressed, many thanks.

5. Comprehensive and well written. In the feeding regimens section, please refer the reader to the supplementary material. I wondered how the protein and energy targets and so on where derived. Eventually I found the info in supplementary material. The information in the supp material is thorough so please encourage the reader to read it.

We have signposted this more clearly in the main text with:

“Further details of the feeding regimes including micronutrients delivered and concentrations of medium-chain triglyceride data can be found in the Online Supplement.”

6. Regarding the section ‘Blood and urine analysis’ - When were samples collected? Was this performed in a consistent and controlled manner? Are these the same samples that were then used for UHPLC-MS metabolomics? Sample collection can have a large effect on the metabolome given the dynamic and unstable nature of plasma metabolites. As much information as possible is required to assess whether sampling may have impacted your results.

We have clarified this further by adding the following into the online supplement, and the section now reads:

“Blood (5 mls in total) and urine (5 mls) were collected in a standardised fashion by research nurses each morning of the 10-day study period. A duplicate of each cryovial was produced.

Whole blood (2mls) was collected into Lithium Heparin tubes. The remaining whole blood was centrifuged at 2500 g for 5 min and supernatant stored at -80°C prior to UHPLC-MS metabolomics analysis.

Blood samples (1.5 mls, in EDTA tubes) and urine samples (in universal tubes) were centrifuged at 2500 g for 5 min and the protein-free supernatant removed and stored at -80°C, before analysis for plasma medium chain fatty acids, pyruvate, lactate and beta-hydroxybutyrate, and urinary beta-hydroxybutyrate, respectively, by gas chromatography/mass spectrometry. For acetoacetate, which is not stable in plasma, 1.5 mls of whole blood was collected at the recruiting site and added to a pre-weighed plastic tube containing 5 ml of 0.77 mol/l perchloric acid previously cooled to 0° C. After mixing, the tubes were re-weighed, centrifuged at 2500 g for 5 min and the protein-free supernatant removed and stored. The dilution of the blood was determined by the weight changes measured. Samples were stored at -80°C until analysis.”

7. Regarding statistical analyses – can you clarify what normalisation and scaling approaches were used?

In the on-line supplement we state “Following recommended guidelines⁸, metabolite features were retained when: peaks were present in at least 70% of pooled QC samples, relative standard deviation was less than 30%, and the extraction blank to mean QC peak area was less than 50%⁸. Probabilistic quotient normalisation was applied to the remaining features, then missing values were imputed using the k-Nearest Numbers algorithm. Data were log transformed prior to analysis.”

We have now added: “

“Autoscaling was performed, where each variable was mean centred and divided by the standard deviation.””

8. Regarding data availability statement - Presumably the metabolite data have been deposited in a repository such as <https://www.ebi.ac.uk/metabolights/> If not, whether the metabolomics analyses planned in the research design? If the analyses were included, what is the reason not gain consent to make the data publicly available?

Metabolomic analysis was pre-planned as per the protocol published on ClinicalTrials.gov. Data deposition has not occurred yet, and will occur following publication. Patients were consented as follows:

“I agree for my anonymised data to be shared with other researchers for further research and research publications on this topic, and for samples to be stored and used in future ethically-approved research”

9. Regarding glucose control, did the authors calculate area under the curve? Might this be an important added insight than just reporting daily variance?

The AUC of plasma glucose was higher in the control arm than in the intervention arm (367.8 vs. 316.9). We have included this and the raw glucose data into the online supplement and refer to the graph in the text.

10. Is it worth even reporting the RF ultrasound scan since the data can't be used?

While we appreciate the statement and the rationale, we have, in keeping with good trial reporting practise, reported these data as per our ClinicalTrials.gov protocol.

11. Are the data from the CPax available and do they show any useful information regarding patient progress?

The median Chelsea Critical Care Physical Assessment score at hospital discharge was higher in the ketogenic feeding arm (34 (95%CI 22-45) vs. 25 (95%CI 8-46).

We have reported this within the manuscript, but have been clear not to do so as an outcome measure, as this is inadequately powered and a post-hoc analysis.

12. Regarding specific metabolite alterations, these metabolites aren't discussed further in the discussion. Could the authors speculate as to the relevance of these key metabolites in driving the changes in pathways.

Thank you for the comment. We have addressed this below (comment 16), where you have asked for a more detailed discussion on metabolites and VIPs

13. In the first paragraph the authors state that 'metabolite abundance data suggest a favourable metabolic profile developed in response to the intervention'. To say this, we need to know what a favourable profile looks like. Can you report what a favourable profile would be? What do healthy controls look like? Are there any published data you could use to support your assertions here?

Thank you for the comment. We have addressed this below (comment 16, 18), where you have asked for a more detailed discussion on metabolites and VIPs, and included literature on ketogenic diets in healthy humans. In addition we have focussed this statement more, which now reads:

"Metabolite abundance data suggest that a favourable metabolic profile developed in response to the intervention, within pathways involved in protein homeostasis and urea cycle flux . This hypothesis requires prospective testing in a larger trial."

14. The authors describe the RQ data as exploratory. Why is it considered so? If the authors hypothesise that the ketones can be effectively absorbed and utilised offering an alternative fuel source in the critically ill, you could also hypothesise that RQ would be lower in ketone supplemented patients.

We considered the RQ data to be exploratory as it was not included in the ClinicalTrials.gov protocol. Further while we agree with the reviewers comment and sentiment about lower RQs we were unable to perform IC in all patients (due to equipment availability and degree respiratory failure (high FiO₂ or PEEP precluding measurements) respiratory failure) so these data are limited by the small numbers.

15. Regarding the discussion of lactate, the control group also showed a decline over time and whilst absolute values differ, relative to baseline it seems like there isn't much difference between groups. I agree that there does seem to be a switch in substrate use but not sure the lactate data support this claim. Could you expand on this and perhaps plot the data relative to the first timepoint for each treatment arm?

Thank you for this comment. Plasma lactate is reflective of a variety of physiological processes (including liver dysfunction and anaerobic respiration) which recover as critical illness is supported and treated, and so the decline in both arms is to be expected. When lactate values are individually normalised to baseline and plotted the difference between arms is no longer apparent (see below). This suggests that the reviewer is correct that these lactate data are not helpful. However we did not specify this analysis in the protocol, and therefore have left the original graph within the text, and modified the discussion to state:

“While the lower lactate levels seen in the intervention arm are an attractive signal of substrate switching, these data are likely the result of baseline variation, and the value of lactate measurements in ketogenic feeding needs to be assessed in a larger cohort”

16. Figure S7 appears to show that the two treatment arms had a fair degree of separation at baseline although the model is perhaps overfitted. Can the authors comment on whether the degree of separation observed at day 10 is due to the treatment received or perhaps partly explained by baseline separation between groups?

In the methods section we state “Models with error rate greater than 20% were considered to be overfitted (i.e., the model described random error in the data rather than relationships between variables)^{1,2}”

With each figure we then show the lack of and real variation respectively for day1 and day 10:

Day 1 : Error rates are >20% in all but one domain: polar positive 17%, non-polar positive 41% , non-polar negative 31%, polar negative 20%, suggesting plot is overfitted and not true variance.

Day 10: All error rates <20% polar positive 12% , non-polar positive 15%, non-polar negative 14%, polar negative 17%. suggesting plot is a result of real variation

Base on the a priori criteria, we can only conclude that although there may seem separation in the baseline, there is too much error to say this is real. However by day 10 the error is within the apriori determined criteria and much reduced, allowing confidence that there is a real difference between the groups.

For clarity, we have placed these statements in the results section in addition to the figure captions:

Sparse Partial Least Squares Discriminant analysis (SPLS-DA) demonstrated no difference in metabolite abundance between arms on Day 1 (*Error rates>20% in all but one domain: polar positive 17%, non-polar positive 41%, non-polar negative 31%, polar negative 20%, suggesting plot is overfitted and not true variance ;Figure S8*). By the end of the intervention period, between-arm differences were seen in 31 non-polar negative and 67 non-polar positive metabolites with Variable Importance in Projection (VIP) scores of >1. Similarly, 45 polar negative and 65 polar positive metabolites had a VIP score >1 (*All error rates <20%: polar positive 12%, non-polar positive 15%, non-polar negative 14%, polar negative 17%, suggesting plot is a result of real variation Figure 3*).

17. Regarding the paragraph on exploratory metabophenotyping - I find this quite weak and it doesn't support the statement made in paragraph 1 of the discussion that 'a favourable metabolic profile developed in response to the intervention'. Are you able to make any comparisons with other studies that report metabophenotyping in critically ill vs healthy controls or indeed just healthy controls? Whilst you report some changes in metabolites that indicate a metabolic shift, can you confidently say that it is towards a 'healthy' metabolite profile.

Whilst the sample is small, the data could be described further. Given VIPs are presented in figure 4 it would be useful to discuss their involvement in important bioenergetic pathways in some more detail. Perhaps comparison to other ketone supplementation studies that have explored pathway alterations? These exploratory type analyses need to give the reader more information on which to build new hypotheses and as it stands, I don't think this section does that very well.

Many thanks for this suggestion which we feel will improve the manuscript. There are few ketone supplementation studies that have used metabolomics, but we reference and compare the existing data from a single study, as requested. This section of the trial is exploratory, and we have tried to strike a balance that is respectful of this. We could expand this section further if required, under editorial guidance:

"Alterations in Cahill cycle flux result in a differential metabolite abundance in the alanine pathway, suggesting a decrease in muscle protein breakdown for alanine production³. This is supported by alterations in Ureidopropionic acid, a urea derivative of beta-alanine and therefore a marker of said

flux. Increase urea cycle flux has been previously described in critically ill patients^{4 5}. Altered alanine and urea cycle flux have also been seen in healthy subjects in response to ketogenic diets though the relevance of this is unclear⁶. Alterations in the abundance of Phosphocholine residue have been noted with ketogenic diets in healthy volunteers, and are hypothesised to be related to carnitine metabolism and mitochondrial fatty acid transport, offering further support to the presence of the uptake and metabolism of decanoic (partially carnitine dependent) acid to maintain ATP production⁶. Lower phosphocholine abundances which may reflect the dysregulation of beta-oxidation have been both described and linked to mortality in critically ill patients⁷⁻⁹. Pentose and glucuronate interconversions suggest a differential regulation of hepatic detoxification potentially related to amino acid breakdown products such as ammonia, and glucuronate abundance differences has been seen previously comparing critically ill trauma patients to healthy volunteers^{10 11}. Ketogenic diets result in altered pentose and glucuronate interconversions in animal models¹². Lithocholate Glucuronide is a bile acid metabolite, which may be related to ketogenic diet related early satiety via Glucagon Like Peptide-1 activation, although differential bile salt abundance was not seen relative to controls in observational studies^{10 13}.”

18. When writing sentences such as ‘Pathways unrelated to metabolism of nutritional lipids were differentially regulated implying tissue metabolism was additionally altered’ please tell us what pathways you’re referring to.

We detail these pathways in the same paragraph, and have now re-structured the paragraph so as to be clearer:

The above paragraph has been substantially re-written.

19. I would also suggest commenting on the utility of untargeted metabolomics in such study designs as yours. The metabolite profiles that you have generated are interesting particularly given the cohort from which they’re derived.

Many thanks for this suggestion. We have added the following:

“Untargeted metabolomic profiling was used in an exploratory fashion, to generate potential panels for future targeted work, coupled with targeted ketone and medium chain fatty acid analyses. The advantage of this approach was seen in the identification of panels separate from those that we would have used a priori e.g. energetic intermediate pathways such as TCA Cycle metabolites and NAD Metabolites. In our future trials, additional targeted panels will include those pertaining to urea cycle flux, glucuronate metabolism, carbohydrate and lipid disposal and ketone metabolites.”

Reviewer #2 (Remarks to the Author):

20. This study implemented a ketogenic diet to critically ill patients with multiorgan failure and

upper level	52.8	61.9	69.9	78.4	86.1	86.4	88.1	87.3	84.3	83.8
lower level	36.0	45.4	54.2	59.9	68.1	66.1	64.1	61.7	63.5	64.2

22. Regarding the secondary endpoints and blood draws, it would be important to know when these samples were collected. Was blood drawn morning, evening, after an overnight fast, after a meal? The timing of the blood draw may influence the metabolomic data.

Patients were fed continuously via nasogastric tube as is standard in the critically ill patient. Blood samples were taken in the morning by research nurses.

23. The patients were reported to have multiorgan failure. Was this confirmed with bloodwork (Liver enzymes, CMP?). Ketogenic diet therapy requires the production of ketones, which is associated with robust hepatic ketogenesis. Reduced hepatic beta oxidation of fats can be a major limitation with this therapy, so it would be important to know this information, and if it influences outcomes.

Organ failure was determined by SOFA scoring, as was pre-specified. This well-validated, extensively researched and implemented score is a mixture of blood tests and clinical physiology. In regards to Liver dysfunction, serum bilirubin informs the SOFA score. The majority of patients in both arms had minimal hepatic dysfunction (median 0, range 0-2). The ketogenic arm data is presented below, with raw bilirubin data and International Normalised Ratio (as a marker of synthetic function) included.

Day	1	2	3	4	5	6	7	8	9	10
Liver SOFA	0 (0-2)	0 (0-2)	0 (0-2)	0 (0-2)	0 (0-2)	0 (0-2)	0 (0-3)	0 (0-3)	0 (0-0)	0 (0-0)
Bilirubin	11 (3-42)	9 (3-40)	9.5 (5-47)	9 (5-61)	8 (4-83)	7 (3-87)	8 (3-109)	6 (3-88)	7 (3-101)	6 (3-110)
INR	1.1 (1.0-1.3)	1.1 (1.0-1.2)	1.0 (1.0-1.3)	1.1 (1.0-1.4)	1.1 (1.0-1.4)	1.1 (1.0-1.46)	1.1 (1.0-1.7)	1.1 (1.1-1.7)	1.1 (1.1-1.9)	1.1 (1.1-1.9)

In the ketogenic arm, one patient had significant hepatic dysfunction (SOFA score >2) and three had scores of >1 for 24-48 hours. Plotting beta-hydroxybutyrate and octanoic acid data of these subjects against the remaining subjects did not reveal an obvious pattern which may be the result of low numbers:

We are unsure of the value of these data in a post-hoc analysis with such low numbers. However we thank the reviewer for this suggestion and have built this analysis into the planned efficacy trial. We would be happy to include this analysis if required under editorial guidance.

24. Line #104: Authors state that in the context of critical illness and organ dysfunction that....."mitochondrial fatty acid oxidation is downregulated, and the resultant inability to use any of these three substrates (glucose, amino acids, fats) efficiently leads to a bioenergetic crisis." This was a rationale for using the ketogenic diet, but this statement contradicts the high fat ketogenic diet approach. The production of ketones is directly dependent upon mitochondrial fat oxidation rates. If the goal is to rapidly induce and sustain therapeutic ketosis as a means to supply alternative energy (BHB and AcAc), why not directly administer dietary ketone bodies or ketone supplements? This should be mentioned in the discussion. Supplemental ketones would circumvent the impairment of mitochondrial fat oxidation, the rate limiting process for "therapeutic ketosis and also circumvent the fat intolerance. Dozens of clinical trials are currently using dietary ketones to circumvent the fat intolerance that is a feature of many chronic illnesses (see clinicaltrials.gov).

Thank you for this insightful comment. We have altered this sentence to read " *peripheral mitochondrial fatty acid oxidation is downregulated*" which is a more accurate representation of the literature, and resolves the contradiction highlighted. In our study, the rationale is that hepatic ketogenesis is coupled with peripheral ketone body utilisation, whereas use of ketone esters (which are artificial and therefore not a supplement) requires hydrolysis of the ketone ester and subsequent peripheral oxidation. We have added the following to the discussion:

"Recently a number of ketone ester products have become commercially available. Ketone esters would circumvent the impairment of mitochondrial fat oxidation, the rate limiting process for therapeutic ketosis and also circumvent the fat intolerance. These products may or may not be feasible to use in critically ill patients and require testing."

Reviewer #3 (Remarks to the Author):

The authors report on the feasibility of using ketogenic enteral feeding in early critical illness. The

paper is well-written with an appropriate study design used to answer the questions. Here are some minor comments.

Many thanks for the supportive comments.

25. In Table 2, some variables are summarized by allocation, using means, while others are summarized using medians. Is there any reason for this? Perhaps it would be worth stating in the methods (Statistical Analyses) that parametric or non-parametric approaches were used to compare means/medians after first checking the distribution of the data? It would be good to mention the variables that were included in the discriminant analysis, and whether there were any issues with missing data and how this was handled.

We have altered the statistical analysis section to:

“Descriptive analysis was performed for the continuous outcomes using mean (95% confidence intervals) and Student’s T-test for parametric data, or median (range) analysed using Mann Whitney U test for non-parametric data”

We have also added the following to the online supplement:

“For PLS-DA all variables are included in the analysis, this being the primary method to reduce variable numbers using the VIP scores. Missing values were imputed after filtering by knn so there were no missing values in the PLS-DA input.

We have also added the missing data percentages:

Day 1	Day 10
HILIC positive - 3.07%	HILIC positive - 3.62%
HILIC negative - 9.19%	HILIC negative - 12.39%
RP positive - 2.72%	RP positive - 2.90%
RP negative - 4.28%	RP negative - 2.58%

26. Under Process and Feasibility of Nutritional Delivery, a snapshot of the questionnaire administered to staff is given in Figure S3. It is not clear whether these are the responses from one of the participants to some of the questions in the questionnaire or not. Giving mean scores to the individual questions as was done for the question ‘How keen would you be to work on another similar study?’ would be more informative.

We have annotated the graph and added the following text to the legend:

“Lack of pharmacists on study meant people were coming to me all day”
 “4 Hour bags were inconvenient and increased work load, 24-hour bag was better”

“Figure S3: Staff questionnaire data on feasibility and acceptability of delivering the intervention Data are median (Inter-Quartile Range) scores for up to n=23. Variation in sample sizes for individual populations come from the multi-disciplinary sample population i.e. not all members of the team were involved in all aspects of the study”

27. Under adverse events on page 9, the "The proportion of patients with diarrhoea was greater in the ketogenic enteral feeding arm (intervention vs. control 76.9% vs. 52.3%)" should read "The proportion of patients with diarrhoea was greater in the ketogenic enteral feeding arm (intervention vs. control 76.9% vs. 53.3%)" as 53.3% is the number in Table S6. Please indicate the N for each group for base excess and bicarbonate in S7.

These have been addressed.

28. Naming of the figures could be simpler, e.g., Figure 2 instead of Figure2ACBD, since the subtitles A to D can be seen on top of the individual figures.

This has been altered as suggested.

29. The labels for Figure 4 are not very clear and could be made bigger in ggplot2.

These labels have been made bigger and clearer.

30. Figure 1AB presents mean (95% CI) of Plasma Beta-hydroxybutyrate and Acetoacetate concentrations during the 10-day intervention. Similar figures are used for other measures (Figure 2, S4 and S5). Because of the small numbers or large standard errors, the lower limit of the CI cannot be shown in these figures. An alternative would be to use jittered boxplots although this might be hard to fit on the graph. The authors are correct to report p-values based on the Mann Whitney-U test in these figures.

Many thanks for considering alternative forms of data visualisation. We have replotted this as a box plot, and feel that the original graph remains superior, and would take editorial guidance here.

Reviewer #4 (Remarks to the Author):

Thank you for the opportunity to review the manuscript: The authors have to be commended for conducting this complex small RCT in challenging times during the COVID-19 pandemic.

Many thanks for the supportive comment.

31. Abstract: Patients receiving ketogenic feeding had fewer hypoglycaemic events (0% vs. 1.58%), required less exogenous insulin (0.0 IU (IQR 0-16) vs. 78 IU (IQR 0-412) but had slightly more daily episodes of diarrhoea (53.5% vs. 42.9%) over the trial period. Why is the 1,58% so detailed while the ketogenic diet is reported a 0% and not as 0,00%? Please provide consistent reporting. Same for 0.0 versus 78. Please provide consistent reporting.

These have been amended

32. It is unclear whether differences are statistically different or only show trends?

We have reported these data in keeping with our published protocol. These data are reporting safety measures only, and we did not *a priori* plan a statistical analysis of data collected.

Introduction:

33. The physiological characteristics of critical illness have significant overlap across a wide range of presenting diseases, challenging commonly used disease-related taxonomies. What is the purpose of the part of the sentence: challenging commonly used disease-related taxonomies? Is this relevant to this paper? Please provide clarification.

Thank you for the question. This statement is relevant in that we hypothesise that the physiological derangements targeted by the intervention are likely to occur across multiple disease states (or taxonomies). Interventions that modify such derangement are therefore likely to have an impact beyond narrow disease taxonomies. We have altered the text to clarify this:

"The physiological characteristics of critical illness have significant overlap across a wide range of presenting diseases, challenging commonly used disease-related taxonomies¹⁴. Interventions that target such derangements are therefore likely to impact on a wide range of diseases"

34. In high intensity exercise, ketogenic diets provide ketone bodies for substrates, improving ATP production decreasing muscle protein breakdown and improving physical performance.¹³ Please add a comma after "ATP production."

This has been added

35. Ketone bodies may therefore offer an alternative substrate source for energy production in critically ill patients. In addition, ketones.. Separate the sentences with a space for clarity.

This has been added.

36. Exclusion criteria: Patients at risk of refeeding syndrome (based on NICE guidelines³¹) were assessed on an individual basis. In several observational studies among ICU patients, refeeding syndrome (RFS) is typically not identified by NICE guidelines but by phosphate drops. Could you explain why NICE criteria were used in a critically ill population?

Thank you for this comment. It is our opinion that there is a difference between refeeding syndrome defined by NICE and the 'refeeding hypophosphataemia' reported in the mentioned observational studies and the mechanisms for the latter remain unknown and unstudied. In the UK, the definition of refeeding syndrome is clear and there are set criteria for assessing patients who are at risk of developing refeeding syndrome (as outlined in the NICE guidelines). The aim of assessing patients for risk of refeeding syndrome is to devise a feeding regimen that aims to prevent this from occurring rather than treating refeeding syndrome after it has occurred (which is what the current ICU studies have reported). As it is common practice in the UK to assess patients for risk of refeeding syndrome

and provide an energy restricted feeding regimen for at least the first 48 hours, we excluded these patients to avoid any confounding from a lower energy delivery.

37. Why was diabetes mellitus (DM) not listed as an exclusion criterion? Please provide an explanation.

Diabetes Mellitus is not a contra-indication to ketogenic diets, and so not listed as an exclusion¹⁵. Use of insulin was not restricted in either arm.

38. Protein targets were individualised to each patient with a range of 0.83 -1.5g/kg/d being used according to specific clinical need. Why were the protein targets set at low levels (0.83 - 1.5g/kg/d)? ESPEN recommends a higher target of > 1.3 g/kg/day. What were the specific conditions for using lower targets in this study?

Thank you for this important question. The aim of this study was not to explore protein delivery, but the feasibility of a ketogenic enteral feeding on inducing ketogenesis. By nature of the ketogenic diet (i.e. the very high fat percentage), it is difficult to meet the high protein targets recommended by the ESPEN guidelines. Further, the evidence to support high protein targets in this population is lacking. Having discussed with several specialist ketogenic dietitians, we felt it appropriate to aim for a minimum level of protein in line with the WHO recommendations, with higher levels being acceptable if the ketogenic feeding regimen enabled this level to be achieved.

39. Patients in the control arm received the site-specific enteral feed as per Trust protocols with an agreed daily energy target to meet their nutritional needs. It appears that the interventions were not isocaloric or isonitrogenous. Please comment on the potential impact of this on the study outcomes.

The reviewer is correct that the interventions were not isocaloric or isonitrogenous. However these would not affect the study outcomes of safety, feasibility, ketogenesis or data collection. Further, as the recent EFFORT-PROTEIN and TARGET trials have demonstrated these relatively minor variations are unlikely to alter outcomes.

40. Multivitamins were administered daily in the ketogenic arm as micronutrients were otherwise not in the modular feed. It seems evident that the ketogenic diet patients received more vitamins than the control patients. Could you provide details on the products used and their dosages? This information is essential as it could introduce a confounding factor affecting the analysis of the ketogenic diet's effect.

We have added the following to the text:

“Control participants received micronutrients as part of their standard feed, and participants in the

intervention group received Sanatogen A-Z, (Bayer, UK1 tablet daily). Further details of the feeding regimes including concentrations of medium-chain triglyceride data (Table S8) and micronutrients (Table S9) delivered can be found in the Online Supplement.”

We are happy to provide the following information on total micronutrient delivery over 10 days, broken down into micronutrient dosing which we have placed in the online supplement :

Micronutrient	Ketogenic Arm	Control Arm
Vitamin A (µgRE)	9060	9379
Vitamin D (µgRE)	50	124
Vitamin E (mg)	110	147
Vitamin C (mg)	600	825
Vitamin B1 (mg)	14	14
Vitamin B2 (mg NE)	16	17
Niacin (mg)	180	324
Vitamin B6 (mg)	20	17
Folic Acid (µg)	2000	2616
Vitamin B12 (µg)	13	29
Biotin (µg)	1500	486
Pantothenic Acid (mg)	60	52
Vitamin K (µg)	200	644
Calcium (mg)	2000	7907
Phosphorus (mg)	1450	5849
Iron (mg)	140	132
Magnesium(mg)	1000	2190
Zinc (mg)	150	115
Iodine (µg)	1500	1337
Copper (µg)	10000	12900
Manganese (mg)	10	27
Chromium (µg)	250	657
Selenium (µg)	550	655

Table 1: Mean total micronutrient delivery over 10 days.

41. Monitoring: Daily rates of high gastric residual volume (GRV>300ml). Why were such low gastric residual volume (GRV) thresholds used (GRV > 300ml)? Typically, abandoning the measurement or setting the threshold at GRV < 500ml is recommended. Please explain the rationale for using a lower threshold.

While we agree that a threshold of GRV<500ml is recommended, standard practise at Barts Health Hospitals was to use a threshold of 300mls at the time of the trial.

42. Sample size calculation: Sample sizes of 12 per arm have been recommended where previous data on which to base a power calculation are lacking³⁴. We aimed to recruit at least 37 patients to allow for a possible high drop-out rate from early death and early recovery, and for protocol violations (common in many critical care trials), and to thus leave 12 patients per arm. What was the basis for considering a dropout rate of 13 out of 37 patients? This dropout rate seems very high, Please provide further clarification on the rationale behind the sample size calculation.

We aimed to recruit at least 37 patients to allow for a possible high drop-out rate from early death and early recovery, and for protocol violations (common in many critical care trials), and to thus leave 12 patients per arm. We have added the following:

“From our previous multi-centre trials and observational studies in critically ill patients with multi-organ failure^{16 17}, drop out levels can be of this magnitude due to early death post-recruitment, unexpected discharge from ICU, extubation earlier than expected, protocol violations and clinical factors (such as a patient not meeting nutritional requirements) leading to the managing clinician transferring them to standard feed.”

Was the inclusion continued until 2x12 eligible and evaluable patients were recruited?

Yes, this is correct. In the event only 29 patients needed to be recruited to achieve 12 eligible and evaluable patients in each arm.

43. Severity of illness:mean (95%CI) APACHE II score was higher in the control arm than the ketogenic feeding arm (21.6 (18.4-24.8) vs. 16.4 (13.5-19.3); p=0.025), admission SOFA scores were similar (9.9 (95%CI8.4-11.4) vs. 10.1 (8.7-11.6); p=0.621). It is uncommon to observe an average APACHE II score of 16.4 in severely ill ICU patients with an average SOFA score of 10.1. Could you please explain these data further and address any potential discrepancies? Are you sure these data are correct? In most studies patients with SOFA scores of 10 and higher have APACHE-II scores > 22.

We approached the national critical care audit in the UK (ICNARC) for their independent APACHE II scores.

This changed the APACHE II data to: ketogenic feeding arm (18.2 (15.5-21.0) versus controls (21.6 (18.4-24.8); p=0.101. Many thanks for identifying a transcription error.

44. Collection of 24-hour urine samples to obtain total nitrogen values was not feasible in the context of heightened infection control during the pandemic. What is the reason for not being able to collect urine during the pandemic? Could you explain the reason why collecting 24-hour urine samples to obtain total nitrogen values was not feasible during the pandemic? Was there any consideration given to collecting smaller samples (from a large volume saved in ICU) or using alternative methods? Please provide more information.

The majority of the data was collected during periods with high numbers of COVID-19 patients in the RLH. This had a major impact on staffing levels, with research nurses being redeployed in front line medical services either on wards or on the ICU. This impacted their ability to collect trial data due to lack of available time given the major clinical workload, and the use of PPE and red/green zoning meant that it was frequently not possible for research nurses to reenter the ICU after finishing their clinical shift. Hence they didn't have control over when they could enter the ICU to collect 24 hour urine samples. Smaller samples were taken (and were used for the urinary beta-hydroxybutyrate levels) but 24 hour sampling is needed for a meaningful estimation of total nitrogen losses.

We did not consider alternative sample collection methods that would have deviated from our protocol.

In response to the reviewers comment, we looked at changes in serum urea and urea-to-creatinine ratios. These biochemical signatures have been shown to reflect changes in both catabolism and patient outcomes after controlling for potential confounders¹⁸⁻²⁰:

Serum urea was lower in the ketogenic arm by day 8 and persisted until the end of the study period (8.7 (95%CI 6.4-19) vs. 12(95%CI 10-19), as was the Urea-to-Creatinine ratio (117 (95%CI 93-206) vs. 168 (95%CI 102-235).

However this is a post-hoc analysis, and was not part of the a priori published trial. We would be happy to include these data under editorial guidance.

45. Variability in glycaemic control improved, and differences between arms in terms of hypoglycaemia, insulin dosing and glucose variability. Typos: hypoglycaemia, should be hypoglycaemia.

This has been altered.

46. While the significance of the equivalence in pyruvate concentrations is unclear, the lower lactate levels seen in the intervention arm are also suggestive of substrate switching. Could it be that the higher acetyl-CoA resulting from the ketogenic diet conversion not only leads to better ATP production but also blocks pyruvate influx, resulting in similar pyruvate levels despite reduced glycolysis? Please comment on this potential mechanism.

Thank you for the excellent suggestion which we have added as a potential mechanism

“the equivalence in pyruvate concentrations despite reduced glycolysis may result from ketone derived acetyl-CoA production leading not only to better ATP production, but also negative feedback on pyruvate influx via pyruvate carboxylase activation²¹.”

47. Alterations in Cahill cycle flux result in a differential metabolite abundance in the alanine pathway, suggesting a decrease in muscle protein breakdown for alanine production. Was the

dietary intake of alanine different between the groups? Please provide additional information and clarification.

Thank you- we have added the following to the results section alongside the VIP data

“Mean nutritional alanine delivery was similar between groups (ketogenic feed 35g (95%CI 25-35) vs. control 27g (95%CI 18-36); p-0.180).”

48. Why is the population in this study 10-12 years younger than typically seen in European ICU studies? This seems unexpected based on the inclusion and exclusion criteria.

The mean age (95% CI) for the group (n=29) is 52.0 (45.5-58.5) years which is similar to that reported in our previous trial recruiting patients with multi-organ failure carried out across 8 ICUs in the UK: mean age (95% CI)(n=121) was 57.7 (54.7-60.6) years.¹⁷ That is quite normal for our deprived local population who have fewer disease free life years compared to the UK and European average.²²

49. Could the effect and feasibility of the intervention be different in a more elderly population? Please discuss the limitations of external validity for the study.

We are unaware of a biological rationale for the efficacy and feasibility to be different as a result of a numerical age. We will be able to explore this in our planned multicentre trial, in which a combination of higher numbers of patients and a widened demographic will allow differences in age response to be examined.

50. For the variables listed in the table, could you provide the corresponding p-values or indicate which variables were statistically significant and which were not? This information would enhance the interpretation of the results.

This has been added

51. Thank you for the opportunity to review this interesting paper.

Many thanks for your review.

REFERENCES

1. Song W, Wang H, Maguire P, et al. Local Partial Least Square classifier in high dimensionality classification. *Neurocomputing* 2017;**234**:126-36.

2. Kelly RS, McGeachie MJ, Lee-Sarwar KA, et al. Partial Least Squares Discriminant Analysis and Bayesian Networks for Metabolomic Prediction of Childhood Asthma. *Metabolites* 2018;**8**(4).
3. Sarabhai T, Roden M. Hungry for your alanine: when liver depends on muscle proteolysis. *The Journal of clinical investigation* 2019;**129**(11):4563-66.
4. Peltz ED, D'Alessandro A, Moore EE, et al. Pathologic metabolism: an exploratory study of the plasma metabolome of critical injury. *J Trauma Acute Care Surg* 2015;**78**(4):742-51.
5. Banoei MM, Vogel HJ, Weljie AM, et al. Plasma metabolomics for the diagnosis and prognosis of H1N1 influenza pneumonia. *Crit Care* 2017;**21**(1):97.
6. Effinger D, Hirschberger S, Yoncheva P, et al. A ketogenic diet substantially reshapes the human metabolome. *Clin Nutr* 2023;**42**(7):1202-12.
7. Ferrario M, Cambiaghi A, Brunelli L, et al. Mortality prediction in patients with severe septic shock: a pilot study using a target metabolomics approach. *Sci Rep* 2016;**6**:20391.
8. Langley RJ, Tsalik EL, van Velkinburgh JC, et al. An integrated clinico-metabolomic model improves prediction of death in sepsis. *Sci Transl Med* 2013;**5**(195):195ra95.
9. Schmerler D, Neugebauer S, Ludewig K, et al. Targeted metabolomics for discrimination of systemic inflammatory disorders in critically ill patients. *J Lipid Res* 2012;**53**(7):1369-75.
10. Parent BA, Seaton M, Sood RF, et al. Use of Metabolomics to Trend Recovery and Therapy After Injury in Critically Ill Trauma Patients. *JAMA Surg* 2016;**151**(7):e160853.
11. Muting D, Kalk JF, Fischer R, et al. Hepatic detoxification and hepatic function in chronic active hepatitis with and without cirrhosis. *Dig Dis Sci* 1988;**33**(1):41-6.
12. Kong C, Yan X, Liu Y, et al. Ketogenic diet alleviates colitis by reduction of colonic group 3 innate lymphoid cells through altering gut microbiome. *Signal Transduct Target Ther* 2021;**6**(1):154.
13. Thomas C, Gioiello A, Noriega L, et al. TGR5-mediated bile acid sensing controls glucose homeostasis. *Cell Metab* 2009;**10**(3):167-77.
14. Maslove DM, Tang B, Shankar-Hari M, et al. Redefining critical illness. *Nat Med* 2022;**28**(6):1141-48.
15. Bolla AM, Caretto A, Laurenzi A, et al. Low-Carb and Ketogenic Diets in Type 1 and Type 2 Diabetes. *Nutrients* 2019;**11**(5).
16. Puthuchery ZA, Rawal J, McPhail M, et al. Acute skeletal muscle wasting in critical illness. *JAMA* 2013;**310**(15):1591-600.
17. McNelly AS, Bear DE, Connolly BA, et al. Effect of Intermittent or Continuous Feed on Muscle Wasting in Critical Illness: A Phase 2 Clinical Trial. *Chest* 2020;**158**(1):183-94.
18. Flower L, Haines RW, McNelly A, et al. Effect of intermittent or continuous feeding and amino acid concentration on urea-to-creatinine ratio in critical illness. *Jpen* 2022;**46**(4):789-97.
19. Haines RW, Zolfaghari P, Wan Y, et al. Elevated urea-to-creatinine ratio provides a biochemical signature of muscle catabolism and persistent critical illness after major trauma. *Intensive care medicine* 2019;**45**(12):1718-31.
20. Haines RW, Fowler AJ, Wan YI, et al. Catabolism in Critical Illness: A Reanalysis of the REducing Deaths due to OXidative Stress (REDOXS) Trial. *Crit Care Med* 2022.
21. Adina-Zada A, Zeczycki TN, Attwood PV. Regulation of the structure and activity of pyruvate carboxylase by acetyl CoA. *Arch Biochem Biophys* 2012;**519**(2):118-30.
22. Wan YI, Robbins AJ, Apea VJ, et al. Ethnicity and acute hospital admissions: Multi-center analysis of routine hospital data. *EClinicalMedicine* 2021;**39**:101077.

REVIEWERS' COMMENTS

Reviewer #1 (Remarks to the Author):

Following the first round of reviews, the authors have made a clear and considered effort to address all of the points that were raised by this reviewer. I think the manuscript is improved and I have no further comments.

Reviewer #2 (Remarks to the Author):

The revised manuscript of original research is considerably improved and presents noteworthy results that will be of high interest to those working in the field of ketogenic therapies.

The revisions and clarification of the methods, analysis and conclusions have considerably strengthened the manuscript and provide a foundation for further studies to advance the science and application of this metabolic-based therapy for critically ill patient management.

Reviewer #3 (Remarks to the Author):

The authors have addressed all the comments I made and I am happy to see the submission proceed to the next stage. On comment number 30, "30. ...An alternative would be to use jittered boxplots although this might be hard to fit on the graph. The authors are correct to report p-values based on the Mann Whitney-U test in these figures." The boxplots look worse than the original plots. It would be better to present the figure in the original format.

Reviewer #4 (Remarks to the Author):

All my comments have been adequately addressed and the manuscript has been improved accordingly. No further comments.

REPLY TO REVIEWERS 2: NCOMMS-23-11933A

Reviewer #1 (Remarks to the Author):

Following the first round of reviews, the authors have made a clear and considered effort to address all of the points that were raised by this reviewer. I think the manuscript is improved and I have no further comments.

Many thanks.

Reviewer #2 (Remarks to the Author):

The revised manuscript of original research is considerably improved and presents noteworthy results that will be of high interest to those working in the field of ketogenic therapies.

The revisions and clarification of the methods, analysis and conclusions have considerably strengthened the manuscript and provide a foundation for further studies to advance the science and application of this metabolic-based therapy for critically ill patient management.

Many thanks.

Reviewer #3 (Remarks to the Author):

The authors have addressed all the comments I made and I am happy to see the submission proceed to the next stage. On comment number 30, "30. ...An alternative would be to use jittered boxplots although this might be hard to fit on the graph. The authors are correct to report p-values based on the Mann Whitney-Utest in these figures." The boxplots look worse than the original plots. It would be better to present the figure in the original format.

Many thanks. We have retained the original format.

Reviewer #4 (Remarks to the Author):

All my comments have been adequately addressed and the manuscript has been improved accordingly. No further comments.

Many thanks.